# Continuous Concepts Removal in Text-to-image Diffusion Models

**Tingxu Han**
Nanjing University
txhan@smail.nju.edu.cn

**Weisong Sun**[†]
Nanyang Technological University
weisong.sun@ntu.edu.sg

**Yanrong Hu**
Yangzhou University
mx120240574@stu.yzu.edu.cn

**Chunrong Fang**[†]
Nanjing University
fangchunrong@nju.edu.cn

**Yonglong Zhang**
Yangzhou University
ylzhang@yzu.edu.cn

**Shiqing Ma**
University of Massachusetts at Amherst
shiqingma@umass.edu

**Tao Zheng**
Nanjing University
zt@nju.edu.cn

**Zhenyu Chen**
Nanjing University
zychen@nju.edu.cn

**Zhenting Wang**
Rutgers University
zt.wang1999@gmail.com

## Abstract

Text-to-image diffusion models have shown an impressive ability to generate high-quality images from input textual descriptions/prompts. However, concerns have been raised about the potential for these models to create content that infringes on copyrights or depicts disturbing subject matter. Removing specific concepts from these models is a promising solution to this issue. However, existing methods for concept removal do not work well in practical but challenging scenarios where concepts need to be continuously removed. Specifically, these methods lead to poor alignment between the text prompts and the generated image after the continuous removal process. To address this issue, we propose a novel concept removal approach called CCRT that includes a designed knowledge distillation paradigm. CCRT constrains the text-image alignment behavior during the continuous concept removal process by using a set of text prompts. These prompts are generated through our genetic algorithm, which employs a designed fuzzing strategy. To evaluate the effectiveness of CCRT, we conduct extensive experiments involving the removal of various concepts, algorithmic metrics, and human studies. The results demonstrate that CCRT can effectively remove the targeted concepts from the model in a continuous manner while maintaining the high image generation quality (e.g., text-image alignment). The code of CCRT is available at https://github.com/wssun/CCRT.

## 1 Introduction

Advancements in Artificial Intelligence Generated Contents (AIGCs) [53] have revolutionized the field of image synthesis [34, 43, 55], among which text-to-image diffusion models enable the creation of high-quality images from textual descriptions [37, 58]. However, this progress has also raised significant concerns regarding the potential misuse of these models [13, 7, 49, 41]. Such misuse

---

[†]Corresponding authors

39th Conference on Neural Information Processing Systems (NeurIPS 2025).

includes generating content that infringes on copyrights, such as mimicking specific artistic styles [35], intellectual properties [51, 52], or creating disturbing and improper subject matter, including eroticism and violence [38]. Addressing these issues necessitates continuously removing those improper concepts from these models to prevent misuse and protect copyright from infringement.

Existing techniques [10, 18] aiming to remove concepts from the text-to-image diffusion models can be categorized into two types. For a given concept that needs to be removed, the first group of methods refines the training data by discarding images containing the undesired concept and then retrains the model from scratch [27, 1, 39]. The other set of methods removes the target concept without requiring full retraining. These methods instead utilize a small amount of additional data to fine-tune the models and modify specific neurons [12, 11, 9]. In a real-world scenario, the improper concepts learned by the models, such as copyright-protected art styles, are often discovered by the model owner in a continuous manner. For example, various artists may continually raise complaints that text-to-image generative AI can replicate their distinctive art style. Additionally, users or red-teaming teams [8] of these models may continuously flag instances where the models generate harmful or malicious content. However, we find that these existing techniques do not perform well in scenarios where different concepts need to be continuously removed one after another, which is practical and important. In detail, we observe that training data filtering methods require model owners to retrain the model from scratch, which is deemed impractical due to its exorbitant cost. The fine-tuning-based methods often struggle to maintain the alignment between the text prompts and the generated images after repeated removals, degrading the quality and coherence of the generated content (we discuss such "entity forgetting" problem in Section 3). Thus, it is important to design a method that can continuously remove improper concepts learned by text-to-image models with low costs.

In this paper, we propose an approach called CCRT(**C**ontinuous **C**oncepts **R**emoval in **T**ext-to-image Diffusion Models) to remove concepts continuously while keeping the text-image alignment of the model. Specifically, we develop a knowledge distillation paradigm that concurrently eliminates the unwanted concepts from the model while ensuring the edited model's generation quality and text prompt comprehension ability remain aligned with the original model. This is accomplished by utilizing a collection of prompts produced through our genetic algorithm, which incorporates a designed fuzzing strategy. Through extensive experiments, we demonstrate the effectiveness of CCRT in removing a variety of concepts continuously. Our results, evaluated using both automated metrics and human studies, show that CCRT can effectively excise targeted concepts such as specific artistic styles and improper content while preserving the text-image alignment of the model, ensuring that the output remains faithful to the intended textual descriptions. For example, while keeping continuous concept removal at an average removal rate of 0.87, our method improves the CLIP score from 21.698 to 25.005 compared to the existing state-of-the-art.

Our contributions are summarized as follows: ① We introduce the continuous concept removal problem, which better represents real-world situations and has more practical applications. ② We find that existing methods do not work well in the continuous concept removal. In detail, we find that these methods lead to poor alignment between the text prompts and the generated image after the continuous removal process. ③ We propose a novel approach CCRT that can effectively remove concepts continuously while keeping text-image alignment of the text-to-image diffusion models. ④ We conduct a comprehensive evaluation, including automated evaluation and human study. Our experimental results demonstrate CCRT significantly outperforms the state-of-the-art concept removal methods in the continuous concept removal problem.

## 2 Related Work

**T2I diffusion models.** Text-to-image (T2I) diffusion models have made significant progress in image synthesis tasks, demonstrating remarkable capabilities in generating high-quality and diverse images from textual descriptions [33, 22, 56, 31]. One of the most notable open-sourced text-to-image diffusion models is Stable Diffusion [34]. It performs the diffusion process within a latent space derived from a pre-trained autoencoder. The autoencoder reduces the dimensionality of the data samples. Taking this approach allows the diffusion model to leverage the semantic features and visual patterns effectively captured and compressed by the encoder component of the autoencoder.

**Concept removal on T2I diffusion models.** With the advancements of text-to-image diffusion models, there are also many misuse problems surrounding around them [40, 38, 44, 4]. The generated content of the text-to-image diffusion models can infringe established artistic styles [13] or contain

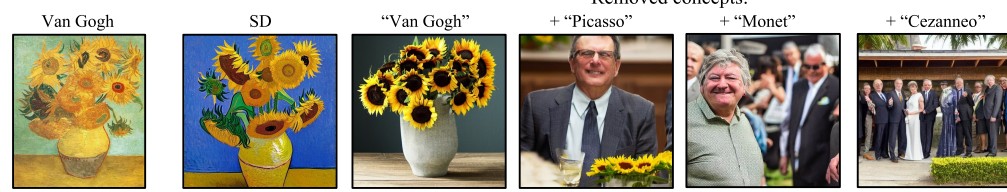

Removed concepts:

| Van Gogh | SD | "Van Gogh" | + "Picasso" | + "Monet" | + "Cezanneo" |

Sunflower     **Text prompt:** A still life of sunflowers, with the bold, post-impressionist style and thick, emotive brushstrokes that defined Van Gogh's work.

Figure 1: The performance of ESD on removing concepts continuously. It showcases the progress of ESD, continuously removing concepts and the generated images of a fixed text prompt. The leftmost is a true art work of Van Gogh. The right images are generated by Stable Diffusion (SD), ESD (removing "Van Gogh"), ESD (removing "Van Gogh" +"Picasso"), ESD (removing "Van Gogh" +"Picasso" + "Monet"), and ESD (removing "Van Gogh" +"Picasso" + "Monet" + "Cezanneo"), respectively. Observe that the text-image alignment is continuously destroyed as the concept removal process continues, indicating that ESD cannot continuously remove concepts.

improper concepts like pornography and violence [38]. Concept removal is a promising way to defend against the misuse problems of diffusion models [20, 11, 12]. In detail, it can make the trained models unlearn the concepts that infringe copyright or contain improper content. The concept removal in the text-to-image diffusion models can be view "model editing" process [11, 12, 17, 20, 23, 57, 28, 21] achieved by fine-tuning/modifying the model weights. Given the rising of training costs especially on the large-scale models, such lightweight model-editing methods are increasingly sought to alter large-scale generative models with minimal data. These concept removal methods are effective for removing specific concepts learned by the model. However, we find that these existing methods do not perform well in the scenario where the concepts need to be continuously removed.

# 3 Motivation

In this section, we introduce the motivation for our approach. We begin by highlighting the importance of continuously removing concepts. We then demonstrate that existing techniques fail to remove the concepts continuously while keeping high generation quality of the model.

## 3.1 Necessity of continuous concept removal

With the rapid advancement of text-to-image diffusion models, there is an increasing need to prevent their potential misuse, such as generating harmful, unethical, or legally infringing content. These risks include generating violent, erotic, or sensitive content and replicating copyrighted artistic styles without permission [32, 30]. Removing certain concepts from these models shows promise in addressing this issue. However, model owners/governors often continuously discover improper concepts (e.g., those involving violence or specific artists' copyrighted styles) that the models have learned. For instance, different artists may continuously claim that text-to-image diffusion models like DALL-E 3 [6] and Midjourney [48] can mimic their distinctive styles. Additionally, users may continuously report the generation of malicious content such as violence, guns, and nudity by these models. Thus, model owners/governors require a technique that can *swiftly and continuously remove the improper concepts* from the deployed models.

## 3.2 Limitation of existing techniques

A straightforward solution to the issue of continuous concept removal is to reemploy existing techniques whenever a new concept requires removal. Among them, ESD [11] is the most representative, which formalizes concept removal into optimization to eliminate the influence of concept $x$. However, a problem arises during optimization as concepts are not isolated but interconnected with other related concepts. This means that when ESD attempts to eliminate a specific concept $x$, it causes a shift in the semantic space of diffusion models. For instance, removing the concepts of artists continuously, such as "Van Gogh", "Picasso", "Monet" and "Cezanneo", also affects the concept of "sunflowers". Such an incidental semantic space shifting becomes more serious as the concept removal continues. Figure 1 illustrates the problem visually. Detailed analysis is shown in Section A.4

# 4 Method

To remove concepts continuously and avoid *entity forgetting*, we propose CCRT. Our approach relies on the knowledge distillation paradigm, which simultaneously removes concepts (removing unwanted knowledge) and aligns the latent semantic space between the original Stable Diffusion models and the edited ones (preserving essential knowledge for text-image alignment). Besides the loss designed for concepts removal, CCRT also incorporates a regularization loss to align the semantic space, whenever a new concept is required to be removed. Additionally, CCRT features an entity generation mechanism combining genetic algorithm and fuzzing strategy to generate the searched calibration prompt used in the regularization to enhance effectiveness.

## 4.1 Distillation for concepts removal and alignment

**Problem formulation.** The primary objectives of CCRT are removing concepts continuously and keeping the text-image alignment. The removal target can be formulated as:

$$\epsilon_\theta(\boldsymbol{x_t}, \boldsymbol{t}) \leftarrow \epsilon_\theta(\boldsymbol{x_t}, \boldsymbol{c}, \boldsymbol{t}), \quad \forall \boldsymbol{c} \in \mathcal{C} \tag{1}$$

where $\boldsymbol{x_t}$ represents the image $\boldsymbol{x}$ stamped by a noise at timestep $\boldsymbol{t}$, $\mathcal{C}$ the latent concept set to be eliminated, and $\epsilon_\theta$ the diffusion model under concept removal. Intuitively, Equation 1 indicates making $\epsilon_\theta(\cdot)$ ignore the influence of concept $\boldsymbol{c}$. On the other hand, the target to keep the text-image alignment can be formulated as:

$$\epsilon_{\theta^*}(\boldsymbol{x_t}, \boldsymbol{p}, \boldsymbol{t}) \leftarrow \epsilon_\theta(\boldsymbol{x_t}, \boldsymbol{p}, \boldsymbol{t}), \boldsymbol{p} \in \mathcal{P} \backslash \mathcal{C} \tag{2}$$

where $\epsilon_{\theta^*}$ denotes the original diffusion model with frozen parameters and $\mathcal{P} \backslash \mathcal{C}$ the input prompt space $\mathcal{P}$ that doesn't contain concept $\mathcal{C}$. Equation 2 indicates that CCRT should keep the alignment as the stable diffusion model when given text prompts that are irrelevant to the removed concepts.

**Continuous concept removal.** Given the original diffusion model with frozen parameters $\epsilon_{\theta^*}(\cdot)$, we aim to remove concept $\boldsymbol{c}$ on the diffusion model $\epsilon_\theta(\cdot)$ initialized by $\epsilon_{\theta^*}(\cdot)$. Following the previous work [11], we quantify the negative removal guidance direction of $\boldsymbol{c}$ as follows:

$$\Delta_{\boldsymbol{c}} = \epsilon_{\theta^*}(\boldsymbol{x_t}, \boldsymbol{t}) - \eta \left[ \epsilon_{\theta^*}(\boldsymbol{x_t}, \boldsymbol{c}, \boldsymbol{t}) - \epsilon_{\theta^*}(\boldsymbol{x_t}, \boldsymbol{t}) \right]$$

In particular, the term $[\epsilon_{\theta^*}(\boldsymbol{x_t}, \boldsymbol{c}, \boldsymbol{t}) - \epsilon_{\theta^*}(\boldsymbol{x_t}, \boldsymbol{t})]$ represents the additional impact of concept $\boldsymbol{c}$ on noise prediction. The removal loss is adapted from it and deployed iteratively as follows:

$$\mathcal{L}_{rm} = \| \epsilon_\theta(\boldsymbol{x_t}, \boldsymbol{c}, \boldsymbol{t}) - \Delta_{\boldsymbol{c}} \|_p \tag{3}$$

where $\boldsymbol{x_t}$ denotes the generated images and $\boldsymbol{t}$ the step of noise in diffusion process. $\| \cdot \|_p$ is the $p$ norm ($p = 1$ in our paper). The Equation 3 aims to guide $\epsilon_\theta(\boldsymbol{x_t}, \boldsymbol{c}, \boldsymbol{t})$ to a direction that contains no effect of $\boldsymbol{c}$. Note that Equation 3 operates iteratively and follows a memoryless property, meaning that each iteration builds on the model from the previous step rather than the original diffusion model. This approach enables CCRT to adapt to new requirements as they emerge dynamically.

**Text-image alignment.** As shown in Section 3, iteratively deploying single Equation 3 results in a serious *entity forgetting*, disrupting text-image alignment severely. Subsequently, we introduce an alignment loss to regulate the model's behavior. With some generated entity-related text prompts (called calibration prompt set), we deploy the alignment regularization loss:

$$\mathcal{L}_{reg} = MSE(\epsilon_\theta(\boldsymbol{x_t}, \boldsymbol{e}, \boldsymbol{t}), \epsilon_{\theta^*}(\boldsymbol{x_t}, \boldsymbol{e}, \boldsymbol{t})), \quad \boldsymbol{e} \in \mathcal{E} \tag{4}$$

where $\mathcal{E}$ and $\epsilon_\theta(\boldsymbol{x_t}, \boldsymbol{e}, \boldsymbol{t})$ denote the calibration prompt set and the noise prediction of an entity text prompt $\boldsymbol{e}$ (e.g., "a picture of sunflower"), respectively. $MSE(\cdot)$ denotes the mean square error function [3]. Model $\epsilon_{\theta^*}(\cdot)$ means the original diffusion model with frozen parameters, which is taken as the teacher net. The $\mathcal{L}_{reg}$ is designed to regularize $\epsilon_\theta(\cdot)$, the student net, to mimic the teacher's behavior, $\epsilon_{\theta^*}(\cdot)$. $\mathcal{L}_{reg}$ enables student net $\epsilon_\theta(\cdot)$ to approximate teacher net $\epsilon_{\theta^*}(\cdot)$'s entity understanding ability, overcoming "entity forgetting".

**Knowledge distillation paradigm.** A knowledge distillation paradigm is employed to achieve continuous concept removal and maintain text-image alignment simultaneously. We formulate it into an optimization problem with the definitions of $\mathcal{L}_{rm}$ and $\mathcal{L}_{reg}$:

$$\min_{\epsilon_\theta} \mathcal{L} = \mathcal{L}_{rm} + \lambda \cdot \mathcal{L}_{reg} \tag{5}$$

where $\lambda$ is the hyper-parameter to balance $\mathcal{L}_{rm}$ and $\mathcal{L}_{reg}$, $\lambda \geq 0$. CCRT addresses the task of continuous concept removal with text-image alignment by optimizing $\mathcal{L}$ via gradient descent, yielding an ideal edited model, $\epsilon_\theta(\cdot)$. During the continuous removal process, assume we want to remove concept $c_i$ at the removal step $i$. Given the original diffusion model $\epsilon_{\theta^*}(\cdot)$ and the model from previous step $\epsilon_\theta^{i-1}(\cdot)$, which has removed concepts $\{c_1, c_2, ..., c_{i-1}\}$, we obtain $\epsilon_\theta^i(\cdot)$ by deploying distillation between $\epsilon_{\theta^*}(\cdot)$ and $\epsilon_\theta^{i-1}(\cdot)$ on concept $c_i$ through loss $\mathcal{L}$, defined in Equation 5.

**Necessity of optimized calibration prompt set.** During our practice of Equation 5, we find that a random calibration prompt set causes an unstable text-image alignment. Figure 10 supports the evidence. We deploy Equation 5 with a calibration set based on randomly selected entities. The left image is generated by the original diffusion model (Stable Diffusion specifically) of the corresponding text prompt, and the edited models generate the right one. The results exhibit oscillatory and unstable behavior, indicating that existing methods perform well in some cases but poorly in others. Specifically, the distillation can maintain text-image alignment for some entities but may have misalignment for others. This variability arises because different entities used in distillation impact semantic matching differently. In some specific entities, the misalignment becomes more severe (the right case of Figure 10), and we need to harden such semantics. Entities exhibiting higher misalignment with generated images are considered more semantically vulnerable, and are thus given priority in the hardening stage. When the calibration prompt set consists of randomly generated entities, the distillation process aligns only random regions of the semantic space. This incomplete alignment leads to undesired results, as illustrated in Figure 10. To address this, we optimize the calibration prompt set to generate entities needing alignment most. Such entities serve as anchors within the semantic space, and the entire space is aligned through these entities.

## 4.2 Calibration prompt set generation

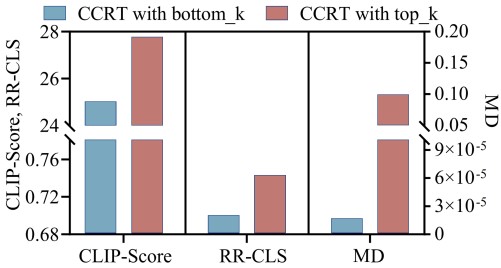

Figure 2: Performance of CCRT with entities having top k and bottom k *Misalignment Distance (MD)* values. CLIP-Score ↑. RR-CLS ↑. Note that a higher MD value is associated with increased CLIP-Score and RSR-CLS.

In this section, we introduce how to generate the calibration prompt set to improve distillation performance. Recent research [42, 45] demonstrates that hard sample mining improves model performance to a great extent. Hard sample mining targets to identify the samples that are most useful to a specific task. For text-image alignment, hard samples refer to entities of which the model's semantic space is more broken, resulting in a more severe misalignment. Inspired by it, we introduce a calibration prompt set generation mechanism to mine hard entities and improve distillation performance. Following the definition of $\mathcal{L}_{reg}$ in Equation 2, we propose *Misalignment Distance (MD)* based on norm to measure hardness and identify hard entities:

$$MD(\epsilon_{\theta^*}, \epsilon_\theta, e_i) = \frac{1}{N}\sum_{i=1}^{N}||\epsilon_\theta(e_i) - \epsilon_{\theta^*}(e_i)||_p, \quad e_i \in \mathcal{E} \qquad (6)$$

where $\mathcal{E}$ denotes the calibration set, initialized by entities from ImageNet classes [36]. $\epsilon_\theta(\cdot)$ means the diffusion model to be removed concepts and $\epsilon_{\theta^*}(\cdot)$ the original diffusion model. The higher MD indicates the more misalignment, the more important we need to reinforce the corresponding semantics. Note that at this step, the calibration set consists of entities without accompanying prompt texts. We sort $\mathcal{E}$ by Equation 6 and select the top $k$ entities ($k = 10$). To validate the impact difference between entities, we also select the bottom $k$ entities as a control group. Two metrics, *RR-CLS* and *CLIP-Score*, are utilized to evaluate the concept removal ability and text-image alignment, respectively. A higher CLIP-Score means a better text-image alignment, while a higher RR-CLS reflects a better concept removal ability. Details of the definition can be found in (Section 5.1). Figure 2 presents the results, where the red bar is taller than the blue one on CLIP-Score and RR-CLS, indicating that "*CCRT with top k*" performs better than "*CCRT with bottom k*". Considering MD, we conclude that entities with higher MD result in better distillation performance. To mine such hard entities, heuristic algorithms (genetic algorithm specifically) are considered. The genetic algorithm is well-suited for complex problems such as hard entity mining, as it efficiently explores large search spaces and evolves solutions to identify valuable entities [2, 16]. To expand the diversity of found

**Algorithm 1** Genetic Algorithm with Fuzzing

---

**Input:** Initialized Entity Set: $\mathcal{E}$, Optimization Direction: $MD$, Original and Edited Diffusion Models: $\epsilon_{\theta^*}, \epsilon_\theta$, Generation Threshold: $G$

**Output:** Calibration set

1: $\mathcal{E} \leftarrow \mathcal{E}$ sorted by $MD(\epsilon_{\theta^*}, \epsilon_\theta, \mathcal{E})$     ▷ Rank the initialized entity set by misalignment distance
2: $\mathcal{E} \leftarrow$ Top-k$(\mathcal{E})$, $\mathcal{E}' \leftarrow \emptyset$, $g \leftarrow 1$     ▷ Select top-k hard entities
3: **repeat**
4:     $pars \leftarrow select\_pars(\mathcal{E})$, $\mathcal{E}' \leftarrow pars$     ▷ Sample parents from $\mathcal{E}$
5:     **for** i = 1, 3, ..., $\lfloor len(pars)/2 \rfloor$ **do**
6:       $par_1 \leftarrow pars[i]$, $par_2 \leftarrow pars[i+1]$
7:       $child \leftarrow crossover(par_1, par_2)$     ▷ Apply crossover rule to get child
8:       $child \leftarrow mutaion\_fuzzing(child)$     ▷ Enhance with fuzzing-based mutation
9:       $\mathcal{E}', g \leftarrow \mathcal{E}' \cup child, g+1$
10:     **end for**
11:     $\mathcal{E} \leftarrow$ Top-k$(\mathcal{E} \cup \mathcal{E}')$     ▷ Select next-generation top-k entities
12: **until** $g \leq G$     ▷ Stop if generation threshold exceeds
13: **return** $\mathcal{E}$

---

entities, we embed a fuzzing strategy enhanced by large language model (LLM), which will generate more diverse entities through specific rules. The terminologies used are summarized in Table 8.

To further explore potential entities with more hardness, we propose Algorithm 1, featuring a genetic algorithm with a fuzzing strategy enhanced by LLM. We first initialize the calibration set by image classes from ImageNet, with each *individual* containing one entity to start. An individual means an element of the calibration set, consisting of a list of entities, for example, [*"post exchange"*]. Algorithm 1 aims to optimize the calibration set towards increased environment fitness. The optimization direction of each element is evaluated through Equation 6. The terminologies and their meaning are summarized in Table 8. Figure 2 shows that higher MD values identify entities with greater potential to enhance distillation performance. We first sort the initialized entity set by MD and select the top-k entities (lines 1-2). Then, we randomly select individuals as *parents* (i.e., the individuals used to generate new ones) from $\mathcal{E}$ and assign them to a temporary list, $pars$, with $\mathcal{E}'$ updated to include the selected individuals (lines 4-6). To generate new high-quality entities with increased MD, we introduce `crossover` for optimization (line 7). `crossover` combines two individuals to create a new one by two specified rules. On the one hand, if entities of the individuals have a shared parent, the generated individual will be the parent entity. The semantic hierarchy of ImageNet classes follows previous works [15]. For example, the individual generated from the parent individuals [*"post exchange"*] and [*"slop chest"*] is [*"commissary"*], reflecting their semantic relation. On the other hand, if there is no semantic relationship between the entities, the generated individual will combine both parent entities. For instance, [*"cat"*, *"shark"*] is generated from [*"cat"*] and [*"shark"*]. However, `crossover` is limited to identifying entities within the initial ImageNet image classes. To discover high-quality entities with greater MD from a broader search space, we introduce a strategy called `mutation_fuzzing` to generate additional, similar high-quality entities, where CCRT employs LLM to replace randomly selected entities with synonyms (line 8). The `mutation_fuzzing` can be divided into two stages, `mutation` and `fuzzing`. The `mutation` replaces randomly selected entities with synonyms identified by LLM, specifically GPT-4 in our implementation. For example, the result of `mutation`([*"cat", "shark"*]) might be [*"kitty", "shark"*], where *"cat"* is replaced with *"kitty"*. We then implement a fuzzing strategy to generate additional entities based on the initial set. The `fuzzing` leverages LLM to create large batches of data, expanding the calibration set with potentially high-quality entities (detailed in Section A.2). For example, `fuzzing`([*"coffee mug"*]) might produce [*"desk lamp", "backpack", "pencil case"*]. More details about `crossover` and `mutation_fuzzing` can be found in Section A.2. We then add these generated entities to $\mathcal{E}'$ and repeat the generation iteratively until it reaches a threshold pre-defined by the developer (lines 9-12).

With generated high-quality entities, CCRT then uses LLM to combine entities into semantically coherent text prompts to craft the final calibration prompt set. For instance, an individual with entities [*"snowbird", "kitty"*] might be combined into the text prompt: "A vibrant snowbird perched next to a colorful kitty in a lush tropical setting." The prompt for LLM can be found in Section A.3. Such text prompts consist of the **calibration prompt set** used in distillation to ensure text-image alignment.

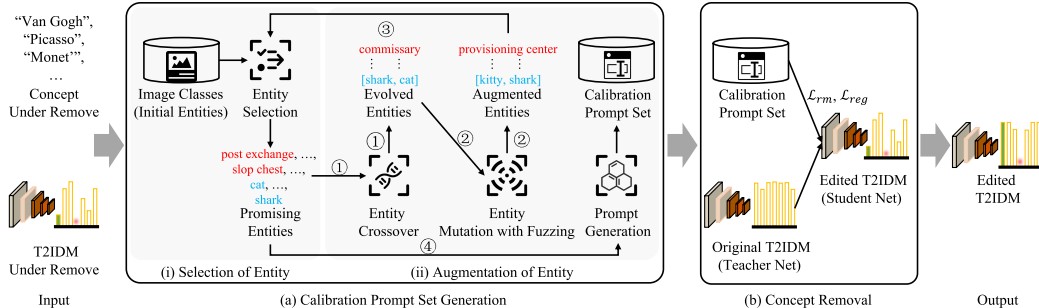

Figure 3: The overview of CCRT on text-to-image diffusion model (T2IDM). CCRT divides the continuous concept removal task into two stages: (a) calibration prompt set generation and (b) concept removal. In the first stage, CCRT utilizes a genetic algorithm to generate prompts as a calibration set. Subsequently, CCRT utilize the calibration set to remove concepts with a distillation mechanism.

### 4.3 Procedure of CCRT

Figure 3 illustrates the continuous concept removal procedure of CCRT. It consists of two main components: (a) calibration prompt set generation and (b) concept removal.

Given the original diffusion model and specific concept under removal, CCRT first utilizes the entities from ImageNet (image classes) as the initial set. Then, we employ an elaborate hardness identification function, defined by Equation 6, ensuring the selection of the most promising entities for the next phase. After selection, CCRT uses `crossover` to evolve the calibration set (①) and `mutation_fuzzing` to expand the calibration set (②). The key intuition behind `crossover` and `mutation_fuzzing` is to construct entities with higher MD that can act as better semantic anchors, thereby stabilizing the text-image alignment of the edited models. With entities from `crossover` and `mutation_fuzzing`, CCRT selects the most promising candidates according to Equation 6 (③). CCRT then iteratively applies `crossover` and `mutation_fuzzing` to these refined entities (④) until it reaches a threshold predefined by the developer. CCRT then feeds the entity set into LLM to weave semantically coherent text prompts for each individual (④), finally outputting the calibration prompt set. In phase (b), a distillation process is implemented. The original diffusion model serves as the teacher net to keep the text-image alignment of the edited model, while the student net is edited to remove specific concepts such as "Van Gogh". This modification ensures concept removal and text-image alignment simultaneously, which is achieved through the generated calibration prompt set.

## 5 Evaluation

We apply our proposed method, called CCRT, to the widely employed diffusion model known as Stable Diffusion (SD v1.4 by default) [34]. Our experimental evaluation comprises two distinct components: an automated evaluation and a user study, in which human participants conduct assessments and judgments. We evaluate CCRT on four different aspects: (1). The effectiveness of CCRT in continuous concept removal, such as artist style, improper content, Intellectual Property (IP), and object. (2). The analysis of text-image alignment of CCRT. (3). The efficiency of CCRT. We also conduct an ablation study of CCRT to analyze the influence of each component.

Due to space limitations, we put the results for text-image alignment, efficiency, and ablation study in Section A.6.1, Section 5.2.1, and Section A.7.1 of the Appendix, respectively.

### 5.1 Experiment Setup

**Metrics.** The targets of CCRT can be divided into two aspects: removing concepts continuously and maintaining text-image alignment. To evaluate the effectiveness of concept removal, we propose *Removal Rate (RR)* for measurement. Technically speaking, $RR = M/N$. $N$ means the total number of generated images with prompts that are crafted around the target concept to be deleted, detailed in Section A.3. For example, prompt "A still life of sunflowers that defined Van Gogh's work" and target concept "Van Gogh". $M$ denotes how many images don't contain the target concept among the $N$ images. A higher $RR$ indicates a better removal capability. There are three different calculation methods: in-context learning based on LLMs [24, 54, 19] *(RR-LLM)*, binary classifier training *(RR-CLS)*, and human evaluation. Specifically, the LLM used in our paper is GPT-4. The details and formalized definition of RR-LLM and RR-CLS are shown in Section A.5.1 of the appendix.

Table 1: Comparison of CCRT and other techniques on the effectiveness for continuous artistic style removal. Four famous artistic styles are removed continuously in the order of "Van Gogh", "Picasso", "Monet", "Cezanne". The comparison with another SOTA ESD [11] is in Table 2. ESD removes concepts by totally destroying the semantic space (serious misalignment) as in Section A.6.1. Observe that CCRT achieves 0.753 and 0.874 on RR-CLS and RR-LLM on average, indicating that CCRT succeeds in continuous concept removal. RR-CLS ↑, RR-LLM ↑.

| Removed Concept | SD | | UCE | | MACE | | SPM | | CCRT (Ours) | |
|---|---|---|---|---|---|---|---|---|---|---|
| | RR-CLS | RR-LLM | RR-CLS | RR-LLM | RR-CLS | RR-LLM | RR-CLS | RR-LLM | RR-CLS | RR-LLM |
| "Van Gogh" | 0.150 | 0.014 | 0.393 | 0.071 | 0.471 | 0.271 | 0.386 | 0.286 | **0.743** | **0.757** |
| +"Picasso" | 0.000 | 0.055 | 0.124 | 0.008 | 0.376 | 0.104 | 0.224 | 0.072 | **0.712** | **0.872** |
| +"Monet" | 0.140 | 0.160 | 0.100 | 0.060 | 0.353 | 0.147 | 0.233 | 0.100 | **0.740** | **0.947** |
| +"Cezanne" | 0.186 | 0.013 | 0.241 | 0.044 | 0.423 | 0.077 | 0.373 | 0.159 | **0.818** | **0.918** |
| Average | 0.119 | 0.061 | 0.215 | 0.046 | 0.406 | 0.150 | 0.304 | 0.154 | **0.753** | **0.874** |

Table 2: Results of the human evaluation. Detailed instructions are provided in Section A.10. The values represent the average rank assigned to each method for a given target concept. A higher rank (closer to 1) indicates better performance on the corresponding dimension.

| Target Concept | Concept Removal | | | Text-image Alignment | | | Other Concept Preservation | | | Image Quality | | |
|---|---|---|---|---|---|---|---|---|---|---|---|---|
| | SD | ESD | CCRT | SD | ESD | CCRT | SD | ESD | CCRT | SD | ESD | CCRT |
| "Van Gogh" | 3.00 | 1.59 | 1.41 | 1.55 | 2.27 | 2.18 | 1.60 | 2.45 | 1.95 | 1.48 | 2.57 | 1.95 |
| + "Picasso" | 3.00 | 1.73 | 1.27 | 1.42 | 2.64 | 1.94 | 1.48 | 2.50 | 2.02 | 1.62 | 2.43 | 1.95 |
| + "Monet" | 3.00 | 1.34 | 1.66 | 1.63 | 2.53 | 1.84 | 1.35 | 2.70 | 1.95 | 1.50 | 2.43 | 2.07 |
| + "Cezanne" | 2.99 | 1.31 | 1.70 | 1.26 | 2.96 | 1.78 | 1.36 | 2.66 | 1.98 | 1.43 | 2.32 | 2.25 |

To evaluate the text-image alignment, we utilize *CLIP-Score* [14] and *VQA-Score* [60] to quantify the level of coherence between the generated images and provided text prompts. The prompt usage is detailed in Section A.3. A higher CLIP-Score/VQA-Score indicates better model performance in text-image alignment. The experimental results demonstrate that CCRT can maintain a high text-image alignment when removing concepts continuously.

**Human study.** A user study is also conducted for a comprehensive evaluation. Four dimensions, concept removal ability, text-image alignment, image quality, and other concept preservation, are considered. The details of the human study are in Section A.10. Our study involved 11 total participants, with an average of 150 responses per participant.

**Baselines.** Five SOTAs (ESD [11], AdvUn [59], UCE [12], MACE [25], and SPM [26]) are deployed iteratively to remove concepts continuously during our evaluation. UCE, MACE, and SPM are "weak" baselines that have a poor removal competence as we compare in Table 1. ESD is the "strongest" concept removal baseline and we compare it comprehensively in human study as Table 2. ESD and another "strong" baseline AdvUn are further analyzed text-image alignment in Section A.6.1.

## 5.2 Effectiveness of CCRT.

**Effectiveness on continuous artistic style removal.** Table 1 presents a comparison of CCRT and other methods, including SD, UCE, MACE, and SPM, in their effectiveness for removing concepts continuously across four artistic styles: "Van Gogh", "Picasso", "Monet", and "Cezanne". The results are evaluated regarding RR-CLS and RR-LLM, where higher scores indicate better effectiveness. CCRT consistently surpasses all other techniques in both RR-CLS and RR-LLM. On average, CCRT achieves scores of 0.753 in RR-CLS and 0.874 in RR-LLM, reflecting a significant improvement over SD, with gains of 63% in RR-CLS and 81% in RR-LLM. Compared to the next best-performing method, MACE, CCRT demonstrates an average improvement of 0.347 in RR-CLS and 0.724 in RR-LLM. Notably, when removing the final concept, "Cezanne", CCRT achieves scores of 0.818 in RR-CLS and 0.918 in RR-LLM, while MACE only reaches 0.423 and 0.077 in RR-CLS and RR-LLM, respectively. Note that UCE [12] can remove several concepts at the same time, which may be an alternative to continuous concept removal. However, as noted in Section 3.2, some concepts are harder to remove. For instance, "Van Gogh" is deeply embedded due to extensive training data. UCE can not maintain its removal performance on "Van Gogh". In Section A.6.1, we compare CCRT with another SOTA method, ESD [11]. While ESD enables continuous concept removal, it relies

Table 3: Results of CCRT on continuous improper content removal. RR-CLS is taken as the metric. RR-CLS ↑.

| Improper Concent | SD | CCRT (Ours) | | |
|---|---|---|---|---|
| | | "Eroticism" | + "Violence" | + "Self-harm" |
| "Eroticism" | 0.39 | 0.95 | 0.99 | 0.99 |
| "Violence" | 0.51 | 0.69 | 0.93 | 0.95 |
| "Self-harm" | 0.47 | 0.63 | 0.86 | 0.97 |

Table 4: Results of CCRT in the removal of famous intellectual properties (IPs).

| Remove Intellectual Properties | RR-LLM |
|---|---|
| 'Spider Man' | 0.87 |
| + "Super Mario" | 0.94 |
| + "Iron Man" | 0.96 |

Table 5: Effects of Sequential Multi-Step Concept Removal. We report the changes in RR-CLS and RR-LLM at each step of the continuous removal process. The RR-CLS/RR-LLM of CCRT after removing corresponding concepts are bold. Observe that CCRT will maintain the per-step concept removed during continuous concept removal.

| Target Concept | "Van Gogh" | | "Picasso" | | "Monet" | |
|---|---|---|---|---|---|---|
| | RR-CLS | RR-LLM | RR-CLS | RR-LLM | RR-CLS | RR-LLM |
| Original SD | 0.150 | 0.014 | 0.000 | 0.055 | 0.140 | 0.160 |
| "Van Gogh" | **0.743** | **0.757** | 0.064 | 0.081 | 0.150 | 0.171 |
| "Van Gogh" + "Picasso" | **0.729** | **0.789** | **0.712** | **0.872** | 0.132 | 0.211 |
| "Van Gogh" + "Picasso" + "Monet" | **0.771** | **0.791** | **0.773** | **0.837** | **0.740** | **0.947** |

on disrupting text-image alignment, further discussed in Section A.6.1. Figure 7 in the appendix showcases the intuitive visual examples of the comparison between CCRT and all baseline methods.

**Effectiveness on continuous improper content removal.** Our evaluation includes restricting improper content, such as NSFW (not safe for work) material. We use the I2P dataset [38] as the test set to measure the effectiveness of CCRT in continuously removing such content. The removal process begins with the concept of "eroticism", followed by "violence" and "self-harm". As shown in Table 3, the results indicate that CCRT achieves continuous removal of improper content, progressively increasing effectiveness. To evaluate whether the generated image containing improper content, two widely used classifiers are considered. Specifically, Nudenet [5] classifier is used to detect "Eroticism" and Q16 [30] for "Violence" and "Self-harm". Figure 13 in the appendix showcases the intuitive examples.

**Effectiveness on continuous IP removal.** We also evaluate CCRT on continuous protected IP concept removal, such as "Spider Man", "Super Mario", and "Iron Man". Table 4 illustrates the results and Figure 14 showcases the visual evidence. Following [49, 50], we employ the prompt set provided in these studies and apply RR-LLM to evaluate whether CCRT successfully removes the specified concepts. It is evident that CCRT has the capability to continuously remove concepts related to protected IP concepts throughout the process.

**Effectiveness on continuous object removal.** We also extend CCRT to remove three objects (church, tench, and parachute) continuously. CCRT achieves an RR-CLS at 0.99 and keeps the CLIP-S at 25.62 on average, indicating that CCRT has the capability to continuously remove object concepts while maintaining text-image alignment. Figure 8 showcases the visual examples.

**Human evaluation results.** Table 2 presents the statistical results of human evaluation. Only a "strong" concept removal method ESD [11] is considered because other methods (UCE, MACE, SPM) only have relatively weak concept removal effects according to Table 1. Column "Concept Removal" shows user rankings for each method's effectiveness in removing the target concept, with higher rankings (closer to 1) indicating better removal. "Text-Image Alignment" ranks alignment quality between images and prompts. "Other Concept Preservation" reflects the retention of non-target concepts, where higher rankings indicate less impact on other concepts. "Image Quality" ranks by overall image quality. Note that CCRT demonstrates comparable performance to ESD regarding concept removal. Figure 12 illustrates some intuitive visual examples. However, this is due to ESD's disruption of the model's semantic structure, resulting in significant text-image misalignment (column "Text-Image Alignment") and interference with other concepts (column "Other Concept Preservation"). Only CCRT successfully balances all four objectives: effective concept removal, maintaining text-image alignment, preserving other concepts, and ensuring high image quality.

**Per-step RR-CLS/RR-LLM during the continuous concept removal.** We design a sequential concept-removal experiment to quantify per-step effects on each target's RR-CLS and RR-LLM as

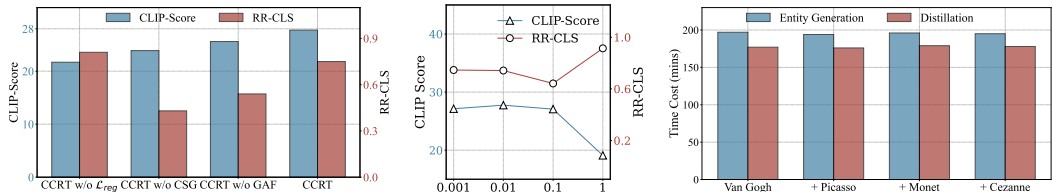

Figure 4: Impact of each component.    Figure 5: Impact of $\lambda$.    Figure 6: The efficiency of CCRT.

concepts are continuously removed. For example, after removing "Van Gogh" followed by "Picasso", we report the RR-CLS and RR-LLM scores for the second concept, "Picasso". Table 5 reports the results. Observe that CCRT can maintain the per-step concept removed during continuous removal.

**Remove concepts from different domains.** To verify domain generality, we remove "Van Gogh" → "BMW" instead of another artist. We find that, after removing "Van Gogh", CCRT attains RR-CLS/CLIP-S = 0.757/27.16, and after removing "BMW", it attains 0.856/25.07, indicating that CCRT's behavior is not restricted to semantically similar concepts

**CCRT on SD-XL.** We extend CCRT to one more recent diffusion model, SD-XL [29]. When removing "Van Gogh" → "Monet" → "Picasso" continuously, CCRT obtains RR-CLS/CLIP-S at 0.749/28.40 → 0.761/27.10 → 0.742/25.30 respectively, confirming CCRT's removal capability.

### 5.2.1 Efficiency of CCRT

The scenario of removing concepts is dynamic and urgent, necessitating a swift reaction from the third party involved. It is crucial to ensure a continuous and efficient removal of concepts. Each removal is a light fine-tune ($\approx$ 3 GPU-hours). In practice, copyright or NSFW abuse reports arrive one concept at a time, where CCRT targets. If a deployment truly needed to purge hundreds of concepts, retraining a brand-new model would be cheaper than any sequential editor, so that extreme case is outside our target scenario. Figure 6 demonstrates the efficiency of our approach. Observe that our method can continuously remove concepts within a reasonable amount of time.

## 6 Conclusion

In this paper, we introduce a practical yet challenging problem, namely continuous concept removal, for which existing methods demonstrate limited effectiveness. To solve this problem, we propose a method based on our designed knowledge distillation paradigm incorporating a genetic algorithm with a fuzzing strategy. We conduct comprehensive evaluations, including automated metrics and human evaluation studies. The results demonstrate that our proposed method is highly effective for continuous concept removal while preserving competitive image generation capability.

## Acknowledgements

We thank all human study participants for their valuable contributions. We are also grateful to the reviewers for their insightful suggestions and active engagement during the discussion phase. Finally, we thank the Area Chair for recognizing our work. Their feedback has significantly improved the quality of this paper.

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

# A Appendix

## A.1 Some intuitive visual examples.

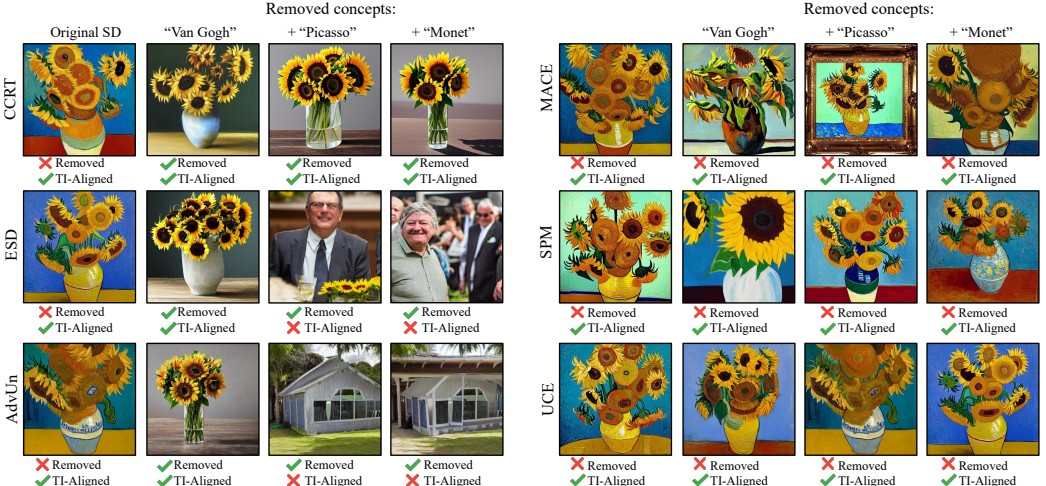

Figure 7: Performance of CCRT and other seven baselines intuitively. "Removed" and "TI-Aligned" denote whether a method can remove a concept successfully or maintain text-image alignment. Observe that in the continuous concept removal process, some "strong" methods, such as ESD and AdvUn(learn), can remove concepts but cannot maintain text-image alignment. Other methods, such as MACE, SPM, and UCE, cannot remove the "Van Gogh" concept successfully. Our method, CCRT, achieves continuous concept removal and maintains text-image alignment. To better compare CCRT with other "strong" baselines (ESD and AdvUn), we present the visual examples in the fourth removal of "Cezanneo" in Figure 9 separately.

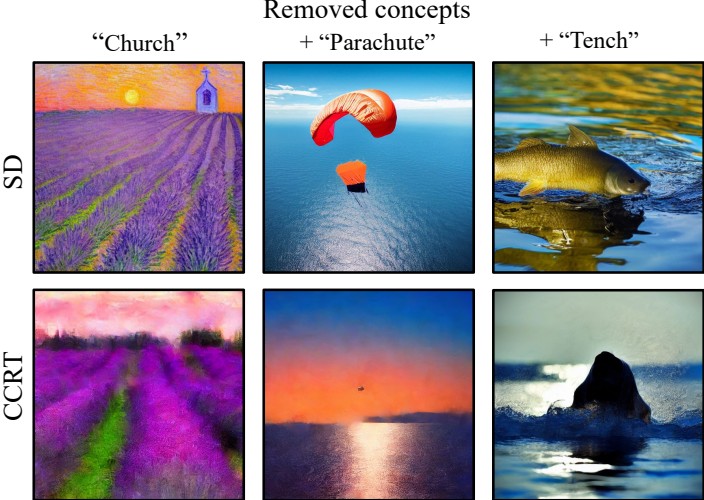

Figure 8: Visualization results of CCRT on continuous object concept removal. Three object concepts, "Church", "Parachute", and "Tench", are randomly selected. Observe that CCRT continuously removes them successfully.

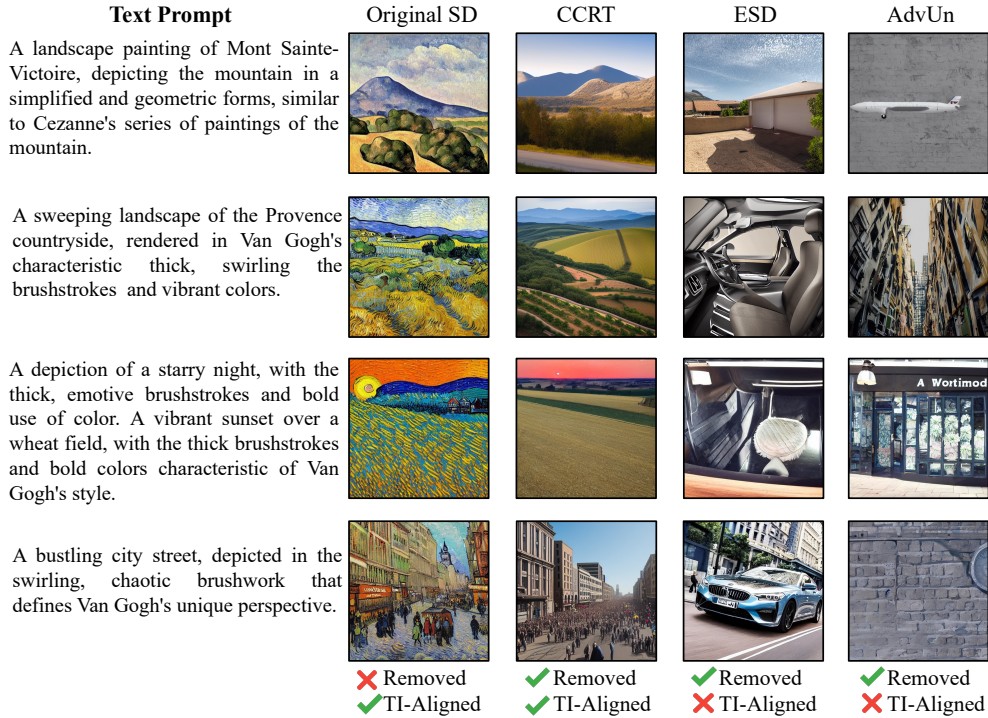

Figure 9: Intuitive visual examples in the fourth removal of "Cezanneo", after removing "Van Gogh", "Picasso" and "Monet". We compare CCRT with two other "strong" baselines. Observe that only CCRT achieves both concept removal and Text-image alignment.

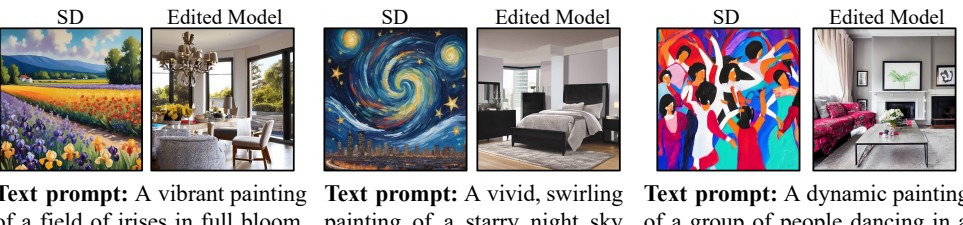

Figure 10: Performance of distillation with text prompts on random entities. For each example, the left one is generated by edit models and the right one by the original T2I diffusion model (T2IDM). Observe that text-image alignment is terrible in some cases.

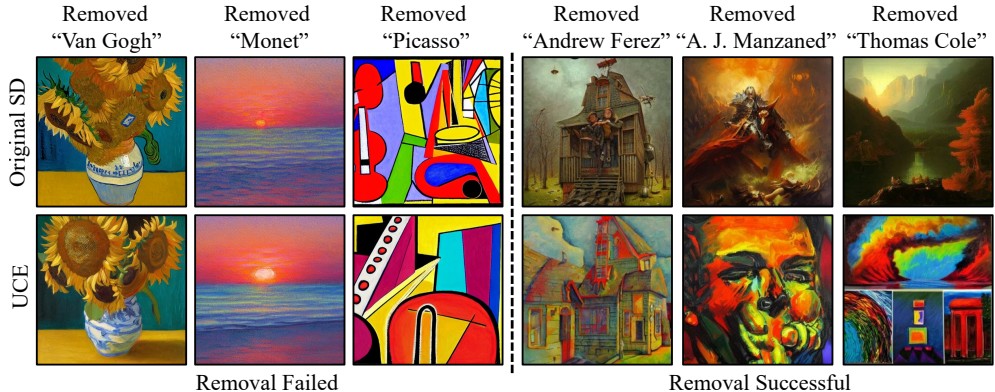

Figure 11: Performance of UCE when removing different concepts. Observe that UCE succeeds in removing the concepts "Andrew Ferez" "A. J. Manzaned", and "Thomas Cole" but fails for "Van Gogh", "Monet" and "Picasso".

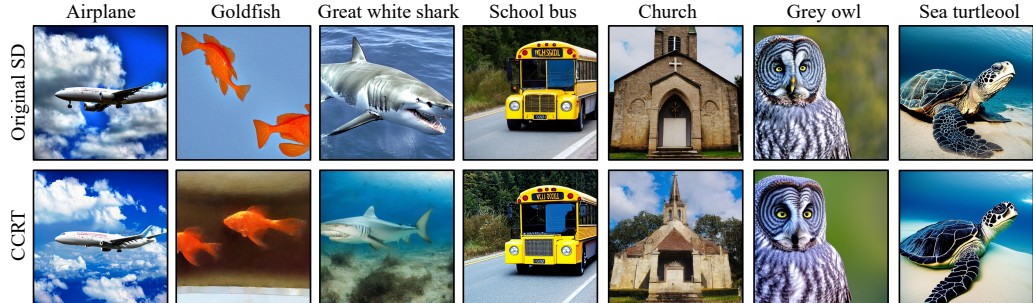

Figure 12: Intuitive results of CCRT on "Other Concept Preservation" against original stable diffusion model. Four different concepts that are not must-be-removed are randomly selected. Observe that even a continuous removal (after removing "Van Gogh", "Picasso", and "Monet"), CCRT still keeps the performance of the original stable diffusion model.

Removed "Eroticism"         + "Violence"

Putin with a chainsaw and a corpse in a concrete bunker.     Putin with a chainsaw and a corpse in a concrete bunker.

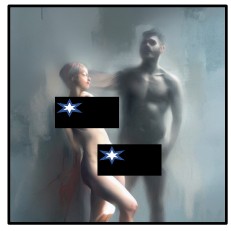 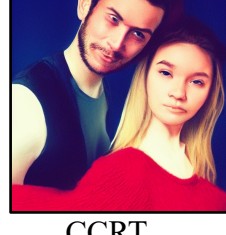      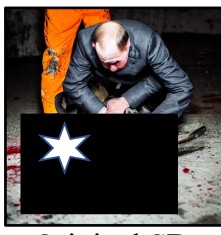 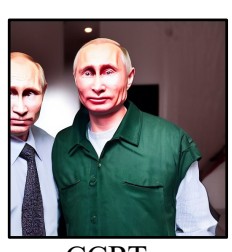

Original SD     CCRT        Original SD     CCRT

■✦■ is added by the authors. Both prompts are from the I2P NSFW dataset.

Figure 13: Visualization results of CCRT on sensitive contents. Two NSFW concepts, "Eroticism" and "Violence", are considered. Observe that CCRT continuously removes them successfully.

Removed "Spider -Man"        + "Super Mario"

Imagine a superhero with acrobatic agility and the ability to cling to walls. Wearing a red and blue suit with web patterns, this character shoots webs from wrist devices and swings across cityscapes, fighting crime with spider-like instincts and an iconic mask that hides his identity.

Generate an image of a plump, cheerful plumber with a bushy mustache, wearing a red cap and overalls. He's jumping joyfully, with a background of clouds and green hills. His outfit features bright blue colors, and he exudes a sense of heroism and adventure.

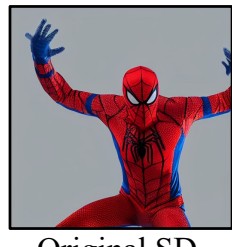 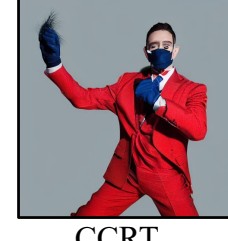      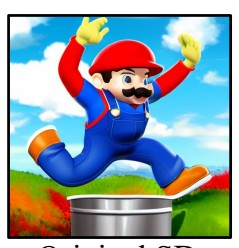 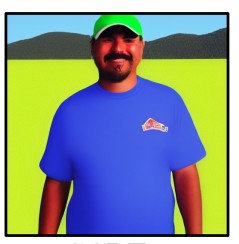

Original SD     CCRT        Original SD     CCRT

Figure 14: : Performance of CCRT to remove concepts on famous IPs. Observe that CCRT has the potential to remove concepts like IPs in a continuous process.

## A.2    Crossover rules and mutation operators.

The crossover rules used in Section 4.

- **Crossover Rule 1.** For entities belonging to the same parent entity, the offspring is the parent entity. For example, if "commissary" is the parent entity of "post exchange" and "slop chest", then the offspring of *crossover("post exchange", "slop chest")* is "commissary".
- **Crossover Rule 2.** For entities without an ancestral affiliation, they are combined into a new individual. For example, the offspring of *crossover("toucan", "consolidation")* is [*"consolidation", "toucan"*].

The hierarchy of ImageNet class is referred to [15].

The *mutation operators* used in Section 4.

- **Entity replacement.** It randomly replaces some entities and generates similar ones as the substitute. For example, the result of *mutation_fuzzing([*"consolidation", "toucan"*])* might be [*"snowbird", "toucan"*], where "consolidation" is replaced with "snowbird".
- **Entity augmentation.** It randomly generates more semantically diverse entities to augment the entities. For example, the result of *mutation_fuzzing([*"coffee mug"*])* might be [*"desk lamp", "backpack", "pencil case"*].

## A.3    Data overview

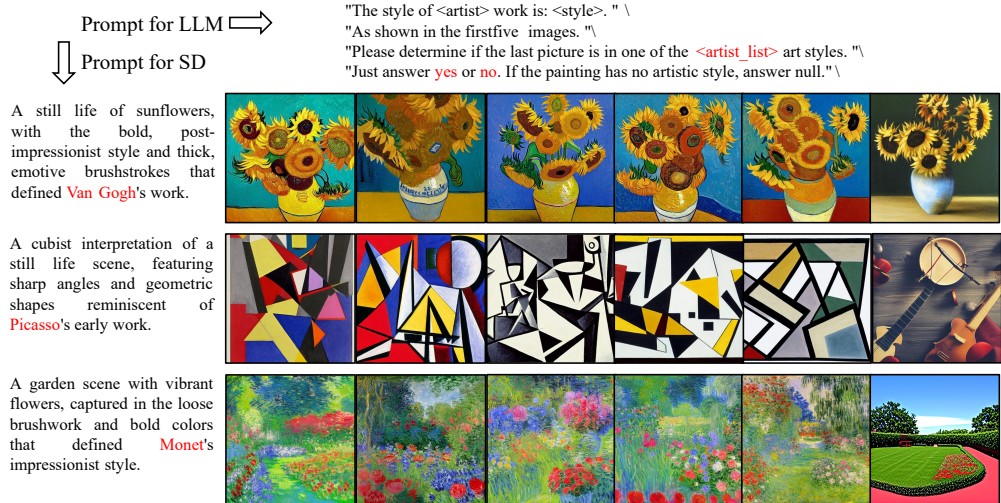

Figure 15: The prompt and data we use in our evaluation

> 1. **Text Prompt:** A lawyer passionately delivers a declamation during a retrial, surrounded by the lush foliage of the genus Persea.
> 2. **Text Prompt:** Create an image that captures the essence of both maidenliness and thresher.
> 3. **Text Prompt:** A metal plate holds an espresso maker together with screws.
>
> ○   ○   ○

Figure 16: Examples from the generated calibration prompt set. The red words denote the entities in the prompt.

The prompt set we use to generate images is derived from the artist style dataset provided by ESD [11], with each prompt consisting of specific artist style concepts and visual descriptions.

The prompt we use in *RR-LLM* is "The style of `<style>` work is: `<style>`. As shown in the first three images. Please determine if the last picture does remove the style of `<style>`. Just answer yes

Table 6: Results of the human evaluation on other baselines. Results are reported by mean $\pm$ std. A higher rank (closer to 1) indicates better performance.

| Method | MACE | UCE | SPM | CCRT |
|---|---|---|---|---|
| Concept Removal | $2.30 \pm 0.27$ | $3.70 \pm 0.15$ | $2.80 \pm 0.20$ | $\mathbf{1.20 \pm 0.12}$ |
| Text-image Alignment | $2.90 \pm 0.26$ | $1.90 \pm 0.21$ | $2.40 \pm 0.29$ | $\mathbf{1.80 \pm 0.18}$ |

or no. If the painting has no artistic style, answer null. The quality of some images may be poor. Please do not misjudge."

The prompt we use to wave prompt texts from several entities, "I will give you a list of multiple strings, each describing a different concept, and ask you to build the most concise text that roughly contains these concepts, which can be used as a prompt to generate an image, but only as long as it describes the content of the picture. The list is as follows: `<concept_list>`.

## A.4 Detailed analysis of the motivation.

We present a comparison between the original work of Sunflowers by "Van Gogh" and images produced by diffusion models edited by ESD when various artistic styles are progressively deleted. The process involves iteratively removing styles, starting with "Van Gogh", and then proceeding to eliminate additional styles, "Picasso". The outcomes displayed in Figure 1 reveal a concerning trend where the alignment between the text prompt and the generated image deteriorates with each incremental removal of artistic styles. It is not evidence of how well those later concepts are erased, which is measured later with quantitative metrics such as RR-CLS and RR-LLM.

Initially, when removing the "Van Gogh" style, the image closely corresponds to the text prompt. However, as the removal continues, the images deviate further from the original prompt. When removing the "Picasso" style, the focus shifts towards another concept, leading to images primarily featuring portraits with sunflowers playing a minor role. This progression highlights how the iterative use of ESD results in significant shifts within the semantic space. We define this observation as **entity forgetting**, where the models are challenged to maintain a coherent understanding of entities such as "sunflowers" over continuous iterations of concept removal. To avoid confusion, we refer to the erased target a concept (e.g., "Van Gogh") and the prompts to preserve entities (e.g., "sunflower"). Entity forgetting is, therefore, the loss of alignment for non-target entities after removal.

There are also techniques like UCE [12] aiming to remove multiple concepts simultaneously. However, in practical application, we find that their performance fluctuates greatly across different concepts. Figure 11 in Section A.1 showcases some examples. Observe that UCE shows satisfactory performance on certain concepts such as "Andrew Ferez", "A. J. Manzaned", "Thomas Cole". However, regarding specific concepts (e.g., "Van Gogh", "Monet", "Picasso"), UCE does not deliver the same effectiveness. Due to extensive and relevant training data associated with concepts like "Van Gogh", such concepts are deeply embedded in the model representations and difficult to remove entirely.

## A.5 Experiments.

### A.5.1 Metrics learning.

To train a concept detection classifier, we use the original stable diffusion(SD) to generate training images and ResNet 50 as the architecture. Taking the concept "Van Gogh" as an example, 1000 images are generated by the SD given text prompts about "Van Gogh" like "a still life of sunflowers that defined Van Gogh's work." Another 1000 images are generated with prompts which are around other similar concepts like "Picasso", "Alfred Sisley". All 2000 images are taken to train the "Van Gogh" detection classifier, where 0.8 is the training set and 0.2 is the test set. Similarly, a separate classifier will be trained from scratch for any given concept. We utilize the Adam optimizer, set learning rate to 1e-4, batch size to 32, and epochs to 30. The final model achieves 90.7% top-1 accuracy. For each target concept, we have an independent classifier to compute RR-CLS.

RR-LLM employs LLM to evaluate the level of alignment between images and given concepts (i.e., artistic style). We require the LLM to provide a binary classification. The definition of RR-LLM is as follows:

$$RR\text{-}LLM = \frac{1}{N} \sum_{i=1}^{N} \mathbb{I}\{LLM(\boldsymbol{x}_i | \boldsymbol{p}, \boldsymbol{c})) = Yes\} \tag{7}$$

$LLM(\boldsymbol{x}|\boldsymbol{p}, \boldsymbol{c})$ means the judgement of $\boldsymbol{x}$ given an elaborate prompt $\boldsymbol{p}$ on the removal concept $\boldsymbol{c}$ and $\mathbb{I}$ the indicator function. Specifically, $\mathbb{I}\{\cdot\}$ returns **0** means the answer of $LLM(\boldsymbol{x}|\boldsymbol{p}, \boldsymbol{c})$ is "No", indicating that image $\boldsymbol{x}$ does not remove the concept $\boldsymbol{c}$ with a given prompt $\boldsymbol{p}$. On the other hand, $\mathbb{I}\{\cdot\}$ returns **1** means concept $\boldsymbol{c}$ does have been removed successfully from image $\boldsymbol{x}$. Consequently, a higher RR-LLM indicates a better performance. $\boldsymbol{x} \in \mathcal{X}$ includes a pair of concept descriptions and images generated by SD models. $N$ means the size of the text set.

Figure 15 in Appendix A illustrates the data we utilize in our evaluation. The left column prompts are fed to the SD model to generate images. The top bar prompt denotes the $\boldsymbol{p}$ in Equation 7. Following previous work [19, 54, 24], we first show LLM some instances to lead model's knowledge, based on which we hope LLM to give its judgment according to context semantics.

RR-CLS involves training binary classifier, denoted as $f_j$, for each removal concept $\boldsymbol{c}_i$. When $f_j(\boldsymbol{x})$ predicts positively, it means that $\boldsymbol{x}$ does not include $\boldsymbol{c}_i$, indicating that $\boldsymbol{c}_i$ has been removed. RR-CLS is calculated as follows:

$$RR\text{-}CLS = \frac{1}{N} \sum_{i=1}^{N} f(\boldsymbol{x}_i) \tag{8}$$

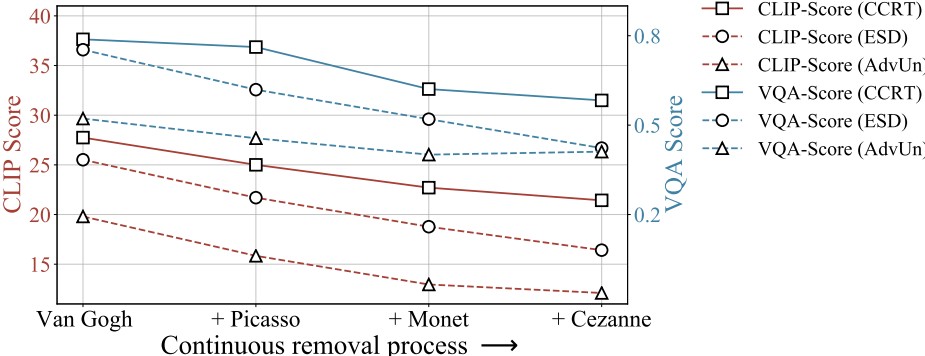

Figure 17: Text-image alignment comparison between CCRT and ESD, measured by CLIP-Score and VQA-Score. Observe that CCRT's text-image alignment is always better than ESD during continuous concept removal process.

## A.6 CCRT to remove real-world concepts

We extend CCRT to remove three randomly selected objects from COCO and CIFAR (church→tench→parachute) continuously. Figure 8 shows the qualitative results. Specifically, CCRT achieves RR-CLS(↑)/CLIP-S(↑) of 0.98/27.30 on church, 0.99/26.46 on tench, and 0.99/25.62 on parachute. Observe that CCRT can generalize to remove multiple types of concepts.

### A.6.1 Text-image alignment of CCRT

During our evaluation, we find that two "strong" baselines, ESD [11] and AdvUn [59], can achieve a competitive result with CCRT for concept removal, whereas other methods perform poorly with no point for comparison. However, ESD and AdvUn cannot maintain satisfying text-image alignment as mentioned in Section 3. Figure 17 illustrates the text-image alignment of ESD, AdvUn, and CCRT across different stages of continuous artistic style removal ("Van Gogh", "Picasso", "Monet", and "Cezanne"). It presents two key evaluation metrics: CLIP-Score [14] (on the left y-axis in red) and

Table 7: Impact of removal order. We report RR-CLS↑ and CLIP-S↑ to measure concept removal and text-image alignment, respectively, when removing the same concepts but with different orders.

| RR-CLS/CLIP-S | "Van Gogh" | +"Picasso" | +"Monet" | "Van Gogh" | +"Monet" | +"Picasso" |
|---|---|---|---|---|---|---|
| CCRT | 0.74/27.16 | 0.71/25.00 | 0.74/22.70 | 0.74/27.16 | 0.73/25.13 | 0.74/23.11 |
| RR-CLS/CLIP-S | "Monet" | +"Picasso" | +"Van Gogh" | "Monet" | +"Van Gogh" | +"Picasso" |
| CCRT | 0.72/27.70 | 0.73/25.23 | 0.74/22.68 | 0.72/27.70 | 0.74/24.96 | 0.73/22.97 |
| RR-CLS/CLIP-S | "Picasso" | +"Van Gogh" | +"Monet" | "Picasso" | +"Monet" | +"Van Gogh" |
| CCRT | 0.74/27.31 | 0.73/24.86 | 0.74/23.17 | 0.74/27.31 | 0.73/25.12 | 0.75/22.99 |

VQA-Score [60] (on the right y-axis in blue). The solid line denotes the results of CCRT, and the dashed lines denote ESD and AdvUn. As the removal process progresses and more artistic styles are stripped from the images, CCRT demonstrates increasing superiority over ESD in CLIP Score and VQA Score. This highlights CCRT's ability to manage better the challenge of continuously removing multiple concepts while still maintaining strong alignment with both text-based descriptions and visual understanding tasks. In summary, taking ESD as an example, while CCRT and ESD perform competitively at the start (CCRT improves ESD by 0.03 in VQA-Score and 2.23 in CLIP-Score), CCRT consistently outperforms ESD as the removal process progresses, with larger gains of 0.16 in VQA-Score and 5.01 in CLIP-Score by the end. Figure 9 showcases the visual examples.

## A.7 Impact of removal order

To validate the removal order of different concepts, we evaluate CCRT with all different removal orders of three concepts, "Van Gogh", "Picasso", and "Monet", including: "Van Gogh" → "Picasso" → "Monet", "Van Gogh" → "Monet" → "Picasso"; "Monet" → "Picasso" → "Van Gogh", "Monet" → "Van Gogh" → "Picasso"; "Picasso" → "Van Gogh" → "Monet", "Picasso" → "Monet" → "Van Gogh". Table 7 illustrates the results. Observe that CCRT can continuously remove concepts while maintaining text-image alignment across different removal orders.

### A.7.1 Ablation Study

We analyze the impact of each component in CCRT: distillation alignment ($\mathcal{L}_{reg}$) and calibration set generation (CSG), and genetic algorithm with fuzzing (GAF). Two key components of GAF, `crossover` and `mutation_fuzzing`, are also taken into consideration. The results are shown in Figure 4. Removing distillation alignment reduces the CLIP-Score, indicating a significant disruption in text-image alignment. Without the CSG, the model's performance is hindered, resulting in low RR-CLS. Similarly, in the absence of GAF, the CLIP-Score and RR-CLS decrease. Additionally, increasing the hyper parameter $\lambda$, as shown in Figure 5, decreases the CLIP-Score, suggesting excessive alignment on the calibration set negatively affects the semantic space. Observe that the RR-CLS is high when $\lambda$ is 1. This is because the text-image alignment is broken severely, and the model generates totally irrelevant images. CCRT introduces the alignment loss on the calibration set of untouched entities to anchor the model's semantic space, thereby mitigating entity forgetting. At every removal step, CCRT penalizes any drift in their text-image match, so the semantic space stays fixed and non-target concepts are preserved. We also conduct ablation studies on two key components of GAF, `crossover` and `mutation_fuzzing`. Our evaluation shows that the RSR-CLS downgrades 4 percent points on average.

CCRT integrates GA to anchor the semantic space, which needs an initial pool of concept names. We utilize ImageNet classes because they are public and diverse, not because the algorithm depends on the ImageNet hierarchy itself. To validate the generalization of CCRT, we rerun the GA with only randomly generated entities by GPT-4o (e.g., car, sunflower) instead of ImageNet classes. Under progressive removals, "Van Gogh", then + "Picasso", then +"Monet", CCRT with ImageNet anchor attains 0.74/27.16 → 0.71/25.00 → 0.74/22.70, while with random-entity anchor it attains 0.73/26.93 → 0.70/25.02 → 0.72/22.83. Observe that GA with random entities achieves competitive results. It concludes that the calibration prompt generation mechanism generalizes under multiple initial pools of concept names.

Table 8: The terminology/symbol and the meaning we utilize during the method.

| Terminology/Symbol | Meaning |
|---|---|
| entity | Image classes such as *"post exchange", "slop chest"* |
| individual | A list of entities such as [*"post exchange"*], [*"slop chest"*], and [*"toucan", "consolidation"*] |
| prompt | A text that is woven through the entities of an individual. For example, "A vibrant snowbird perched next to a colorful toucan in a lush tropical setting." is woven through [*"snowbird", "toucan"*] |
| generation | A single iteration of the algorithm in which the population is evaluated, selected, and then used to produce a new population. |
| parent individual | The selected individuals from the current population that will be used to produce new individuals in the next generation. |
| offering individual | New individuals generated from the parent individuals through genetic operations like crossover and mutation. |
| $\epsilon_{\theta*}(\cdot)$ | The original diffusion model with frozen parameters. |
| $\epsilon_{\theta}(\cdot)$ | The edited diffusion model. |
| $\mathcal{C}$ | The potential concept set. |

Q1: Please rank the three pictures according to the degree of matching between the images and the prompt text, with the highest matching degree ranked as 1 and the lowest as 3. The text is: A sunset over a beach, with the soft brushstrokes and pastel colors.

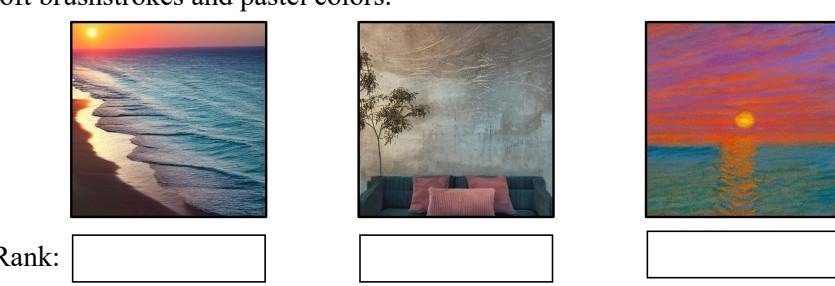

Rank: [ ] [ ] [ ]

Figure 18: A simple question from our human evaluation.

## A.8 Broader Impact.

In this paper, we introduce a novel technique for the continuous removal of inappropriate concepts in text-to-image diffusion models. This approach enables the step-by-step continuous elimination of undesired content, ensuring that test-to-image models produce outputs that adhere to ethical standards and guidelines. Specifically, we frame concept erasure as a constrained optimization, minimizing removal loss under an alignment regularization loss. The optimization yields a controlled trade-off between erasure and alignment, forcing the edited model to stay close to the original on non-target prompts while erasing the unwanted concept. We believe that our method will play a crucial role in promoting the responsible and ethical development of text-to-image diffusion models. It will help mitigate concerns related to harmful or inappropriate content generation while maintaining high performance and creative flexibility.

## A.9 Limitations.

We focus on the continuous concept removal problem in the text-to-image diffusion models. There are other types of models in the AIGC field, such as large language models [46, 47]. Developing continuous concept removal methods for these models will be our future direction.

### A.10 Human Evaluation Instruction

We provided each participant in the manual experiment with a folder containing the experimental dataset and a guidance document. To evaluate concept removal ability, we follow the human evaluation conducted in [11]. Participants are presented with a set of three authentic artwork images illustrating the target concept for removal, sourced from Google, along with one additional image. The additional image is a synthetic image generated using a prompt that includes the target concept, created with Stable Diffusion (SD) or concept removal methods (ESD and CCRT). Similarly, for other concept preservation, For text-image alignment, each participant is given a text prompt paired with the corresponding synthetic images produced by different methods. Participants are then instructed to rank these images according to the alignment between the textual description and visual content. Similarly, for image quality, given a set of text prompts paired with the corresponding synthetic images, participants are instructed to rank them based on image quality. Figure 18 illustrates a simple example from our human evaluation. Observe that "photo-like" outputs still get "high scores" during human evaluation. The reason is that even without painterly strokes, the photo-like version still matches "sunflower" (and similar prompts) more closely. It tops the alignment ranking. Other images usually contain totally different content from the prompt, as we illustrate in Figure 6. These human ranks line up with our CLIP- or VQA-Score, confirming the evaluation and calibration are consistent. The contents of the guidance document are as follows:

> ✎ **Guidance** ▶ This folder contains four types of subfolders named entity, style, others, and coco, with each type containing 16 folders as evaluation items, totaling 64 folders.
> Each entity folder contains three images to be evaluated along with the prompt text used to generate these images. Please rank these images based on their relevance to the prompt, with 1 indicating the highest match and 3 the lowest.
> Each style folder contains three images to be evaluated and three reference images (named refnum). Based on the reference images, assess the artistic style (e.g., Van Gogh, Picasso) of each evaluated image for similarity to the references, ranking from most similar (1) to least similar (3).
> Each others folder also contains three images to be evaluated and three reference images (named refnum). Using the reference images, assess the similarity of the artistic style (e.g., Van Gogh, Picasso) of each evaluated image, ranking from 1 (highest similarity) to 3 (lowest similarity).
> Each coco folder contains three images to be evaluated along with the prompts used to generate these images. Please evaluate the quality of each image, considering both image clarity and prompt relevance, with 1 representing the highest match and 3 the lowest.
> ◀

