# OpenReview forum: "Continuous Concepts Removal in Text-to-image Diffusion Models"
_NeurIPS.cc/2025/Conference — NeurIPS 2025 poster_

### Official Review · Reviewer_XZUm · 2025-06-07

**Clarity:** 2
**Significance:** 3
**Originality:** 2
**Rating:** 3
**Confidence:** 4

**Summary:**

This paper addresses the challenge of continuously removing inappropriate or copyrighted concepts from text-to-image (T2I) diffusion models while maintaining text-image alignment. The authors identify that existing concept removal methods fail in practical scenarios requiring sequential removal of multiple concepts. For example, iterative removal of artistic styles (e.g., Van Gogh, Picasso) using state-of-the-art methods like ESD leads to severe "entity forgetting," where the model loses coherence between text prompts and generated images. To solve this, the authors propose CCRT (Continuous Concepts Removal in T2I Diffusion Models), a knowledge distillation framework combined with a genetic algorithm (GA) and fuzzing strategy. Experimental evaluations on Stable Diffusion demonstrate that CCRT outperforms baselines in continuous removal of artistic styles, improper content (e.g., eroticism, violence), and intellectual properties (e.g., Spider-Man). However, the paper is too verbose to contain more substantial experimental results or method details in the main paper.

Last but not least, the authors show obvious hostility towards Russian President Putin. For example, in Figure 10, Putin is used to show eroticism and violence concepts. However, from my perspective, the academic papers should not convey such a view. The same experimental effect can be completely expressed in other forms without using a well-known person.

**Questions:**

1.	The authors propose to remove semantic concepts continuously one by one. Can some concepts be removed simultaneously? For example, two concepts are removed once.

2.	The concepts are removed continuously one by one. It is necessary to show the impact of the removal order of different concepts.

3.	While evaluating the Effectiveness on continuous artistic style removal and Effectiveness on continuous IP removal, the details of training of the classifier for RR-CLS metric should be provided.

4.	From the visualized results in Figure 10 and Figure 11, we can see that the image quality is far from satisfactory when some concepts are removed.

**Ethical Concerns:**

["NO or VERY MINOR ethics concerns only"]

**Final Justification:**

Thanks for the authors' response. Most of my concerns are solved. However, the visual quality of the generated images produced by SD-1.4 or other duffusion based methods are still doubtful. Based on these, I decide to raise my rating score from 2 to 3.

**Limitations:**

The main experiments are conducted on Stable Diffusion models. However, there are lots of generative models. The proposed method should be applied to other famous models to evaluate the generalization ability of the proposed method.

**Paper Formatting Concerns:**

Some verbose statements can be removed to leave space for the method details and more experimental results.

**Quality:**

2

**Strengths And Weaknesses:**

The task of continuous concept removal is an interesting task, and it is a practical problem in some real-world scenarios. This work can address real-world needs for iterative model editing in response to evolving copyright or ethical concerns to some extent. Nonetheless, there exists some weaknesses.

1.	The main paper refers to the supplementary a lot to describe the proposed method. Which is not suitable. For example, the contents in line 183-198 refer to Figure 7 heavily to describe the necessity of optimized calibration prompt set.

2.	The writing of this paper is too verbose. No substantial experimental results are provided even by page 7, which is not suitable for a NeurIPS paper that is limited to 9 pages.

---

> ### Author Rebuttal · Authors · 2025-07-29
>
> **Response to Reviewer XZUm**
>
> We thank the reviewer for acknowledging the novelty of our method and the strength of our experimental results. We hope the following results and clarifications can address your concerns. We are happy to provide more clarifications and results if needed.
>
> **Q1:** Writing and structure issues. Some results are in the Appendix.
>
> **A1:** Thank you for pointing this out. In the revision, we will
>
> (1) move Figure 7 and its supporting discussion from the Appendix to the main text, immediately near lines 183-198. This change makes the calibration-prompt rationale fully self-contained without forcing readers to flip to the supplementary,
>
> (2) move some detailed comparisons in the motivation to the Appendix,
>
> (3) collect lengthy qualitative galleries and calibration-prompt examples into Appendix,
>
> (4) with the saved space, move the experimental results to the main paper.
>
> **Q3:** Delete multiple concepts at once.
>
> **A3:** Thanks for your suggestion. In the real world, the misuse report of improper concepts usually emerges one at a time, which is the scenario our paper and most current concept removal techniques [1-3] target. Deleting multiple concepts at once is still an open-problem. The effectiveness of existing approaches that attempt simultaneous removal (e.g., UCE[1]) is limited, as we report in Figure 8. For example, when removing concepts such as "Van Gogh", "Picasso", and "Monet" all at once, UCE has a poor removal rate of 0.273, 0.132, and  0.231. Removing multiple concepts at once is beyond the scope of this paper, and we will further explore it in future work.
>
> [1]. Yimeng Zhang, et al. Defensive unlearning with adversarial training for robust concept erasure in diffusion models. NeuralPS 2024.
>
> [2]. Rohit Gandikota, ea al. Unified concept editing in diffusion models. WACV 2024.
>
> [3]. Shilin Lu, et al. Mace: Mass concept erasure in diffusion models. CVPR 2024.
>
> [4]. Rohit Gandikota, et al. Unified concept editing in diffusion models. 2024.
>
> **Q4:** The removal order of different concepts.
>
> **A4:** Thanks for your comment. As suggested, we evaluate CCRT with all different removal orders of three concepts, "Van Gogh", "Picasso", and "Monet", including:
>
>  "Van Gogh" --> "Picasso" --> "Monet", "Van Gogh" --> "Monet" --> "Picasso";
>
>  "Monet" --> "Picasso" --> "Van Gogh", "Monet" --> "Van Gogh" --> "Picasso";
>
>  "Picasso" --> "Van Gogh" --> "Monet", "Picasso" --> "Monet" --> "Van Gogh";
>
>  The table below illustrates the results. Observe that the removal order does not influence CCRT's performance. CCRT can continuously remove concepts while maintaining text-image alignment across different removal orders.
>
> | RR-CLS$\uparrow$/CLIP-S$\uparrow$     | "Van Gogh"     | + "Picasso"      | + "Monet"        |
> | ------------------------------------- | -------------- | ---------------- | ---------------- |
> | CCRT                                  | 0.743/27.16    | 0.712/25.00      | 0.740/22.70      |
> | **RR-CLS$\uparrow$/CLIP-S$\uparrow$** | **"Van Gogh"** | **+ "Monet"**    | **+ "Picasso"**  |
> | CCRT                                  | 0.743/27.16    | 0.731/25.13      | 0.742/23.11      |
> | **RR-CLS$\uparrow$/CLIP-S$\uparrow$** | **"Monet"**    | **+ "Picasso"**  | **+ "Van Gogh"** |
> | CCRT                                  | 0.720/27.70    | 0.738/25.23      | 0.747/22.68      |
> | **RR-CLS$\uparrow$/CLIP-S$\uparrow$** | **"Monet"**    | **+ "Van Gogh"** | **+ "Picasso"**  |
> | CCRT                                  | 0.720/27.70    | 0.741/24.96      | 0.739/22.97      |
> | **RR-CLS$\uparrow$/CLIP-S$\uparrow$** | **"Picasso"**  | **+ "Van Gogh"** | **+ "Monet"**    |
> | CCRT                                  | 0.749/27.31    | 0.737/24.86      | 0.741/23.17      |
> | **RR-CLS$\uparrow$/CLIP-S$\uparrow$** | **"Picasso"**  | **+ "Monet"**    | **+ "Van Gogh"** |
> | CCRT                                  | 0.749/27.31    | 0.732/25.12      | 0.752/22.99      |
>
> **Q5:** Training details of the classifier used for RR-CLS.
>
> **A5:** Thanks for your comment. RR-CLS utilizes ResNet 50 as the backbone. We train it on 2000 images as the training dataset, 1000 for the target concept, and 1000 from other concepts. Adam optimizer, learning rate 1e-4, batch size 32, 30 epochs. The final model reaches 90.7 percent top-1 accuracy. For each target concept, we have an independent classifier to compute RR-CLS. We will add related settings and results in the revised version.
>
> **Q6:** Image quality in Figure 10 and Figure 11.
>
> **A6:** Thanks for pointing it out. The blur comes from the base model SD-1.4, which often fails on fine details at its 512 × 512 resolution and is known to distort hands and faces. We run the same removal on SD-XL, which produces sharp and faithful images. We will update the images of our experiment in the revised version.
>
> [1]. Prompting Pixels. Comparing Stable Diffusion Models. Medium, 2024.
>
> [2].Dustin Podell et al. SDXL: Improving Latent Diffusion Models for High-Resolution Image Synthesis. arXiv:2307.01952, 2023
>
> **Q7:** CCRT on other generative models.
>
> **A7:** In line with previous concept removal techniques [1, 2, 3], we designed CCRT to operate on open-source, publicly available text-to-image diffusion backbones. Stable Diffusion (SD) is by far the most widely adopted model in both research and practice. In contrast, other leading generative models, such as DALL·E 3 and Midjourney, are closed-source and not accessible for model editing. Accordingly, following previous works [1–3], our main experiments use SD v1.4 as the testbed.
>  During the rebuttal, we have now also applied CCRT to a more recent diffusion backbone, SD-XL. The additional results (shown below) confirm that CCRT can continuously remove concepts across models.
>
> | RR-CLS/CLIP-S  | “Van Gogh”  | + “Monet”   | + “Picasso” |
> | -------------- | ----------- | ----------- | ----------- |
> | CCRT[SD v1.4 ] | 0.743/27.16 | 0.712/25.00 | 0.740/22.70 |
> | CCRT[SD-XL]    | 0.749/28.40 | 0.761/27.10 | 0.742/25.30 |
>
> [1]. Yimeng Zhang, et al. Defensive unlearning with adversarial training for robust concept erasure in diffusion models. NeuralPS 2024.
>
> [2]. Rohit Gandikota, ea al. Unified concept editing in diffusion models. WACV 2024.
>
> [3]. Shilin Lu, et al. Mace: Mass concept erasure in diffusion models. CVPR 2024.

---

> ### Author Response · Authors · 2025-08-03
> **A friendly reminder**
>
> Dear Reviewer XZUm,
>
> Thank you again for your valuable questions and the time you've dedicated to our submission. We genuinely hope you could have a look at the new results and clarifications, and kindly let us know if they have addressed your concerns. We would appreciate the opportunity to engage further if needed.
>
> Best,
>
> Authors of Paper 5869

---

> ### Author Response · Authors · 2025-08-05
> **A friendly reminder**
>
> Dear Reviewer XZUm,
>
> Thank you once more for your valuable questions and the time you've dedicated to our submission. We have uploaded additional results and clarifications in response to your comments and would be grateful if you could let us know whether they resolve your concerns. We are happy to provide further details and appreciate the opportunity to engage further if needed.
>
> Best regards,
>
> Authors of Paper 5869

---

> ### Author Response · Authors · 2025-08-06
> **A friendly reminder**
>
> Dear Reviewer XZUm,
>
> Thanks for your previous advice and your precious time on our work. The discussion period will conclude soon, and we would greatly value your thoughts on the additional experiments and clarifications we posted in response to your earlier concerns. Your suggestions are important to us and we appreciate the opportunity to engage in the discussion.
>
> We are also happy to provide further clarifications or experiments if needed.
>
> Thank you again for your time and consideration.
>
> Best regards,
>
> Authors of Paper 5869

---

> > ### Comment · Reviewer_XZUm · 2025-08-07
> >
> > Thank Ac for the reminder. My rating has been updated.

---

### Official Review · Reviewer_xaBb · 2025-07-01

**Clarity:** 2
**Significance:** 2
**Originality:** 3
**Rating:** 4
**Confidence:** 2

**Summary:**

Propose a method that can continuously remove unwanted concepts (such as copyrighted artistic styles, sensitive content, or intellectual property) from text-to-image diffusion models while preserving text-image alignment and generation quality. Contributions are:
1. Formulation of the continuous concept removal problem
2. Demonstration that existing methods' drawbacks
3. Methd explanation and illustration of CCRT
4. Experiments and Evaluation

**Questions:**

Have you evaluated CCRT on other major text-to-image diffusion models beyond Stable Diffusion v1.4 (e.g., SDXL, DALL-E 3, Midjourney)?
What are the computational costs of CCRT, particularly for the calibration prompt generation (genetic algorithm + fuzzing) and the distillation process?
Could you provide more concrete examples of calibration prompts produced by the genetic algorithm + fuzzing?
How critical is the full genetic + fuzzing calibration prompt search?  Any ablation studies?

**Ethical Concerns:**

["NO or VERY MINOR ethics concerns only"]

**Final Justification:**

The rebuttal adds SD-XL results showing consistent performance beyond SD-1.4, reports clear compute costs, and improves deployment realism with real-world targets plus planned case studies. The authors will move key ablations/efficiency to the main text, provide concrete calibration-prompt examples, and include ablations validating the full genetic+fuzzing search; they also commit to a Responsible Use section. Remaining concerns are modest—possible over-suppression of broader painting styles when erasing specific artists, partial reproducibility of the GA+LLM prompt pipeline (seeds/code release), and stronger statistics for the human study. Overall, these updates address my main reservations. I have updated the score correspondingly.

**Limitations:**

The paper does not address how their method could be repurposed to selectively remove concepts in a way that biases or censors model outputs (e.g., erasing cultural, political, or minority representations unfairly).

**Paper Formatting Concerns:**

No majors to report

**Quality:**

2

**Strengths And Weaknesses:**

1. Proposes a novel, well-designed method (distillation + optimized calibration prompts).
2. Strong results: better removal and alignment than prior work.

Weaknesses:
1. Tested mainly on Stable Diffusion; lacks model diversity.
2. Computational cost not reported.
3. No real-world deployment examples
4. Some clarity issues; key details in appendix.

---

> ### Author Rebuttal · Authors · 2025-07-29
>
> **Response to Reviewer XaBb**
>
> Thank you very much for your constructive comments and recognition of the originality of our work. We hope the following results and clarifications can address your concerns. We are happy to conduct more experiments and provide more clarifications if needed.
>
> **Q1:** CCRT on models beyond Stable Diffusion v1.4 (e.g., SD-XL, DALL-E 3, Midjourney)?
>
> **A1:** Thank you for the suggestion. Following previous works [1, 2, 3],  CCRT is white-box and removes concepts continuously through model editing. Because DALL·E 3 and Midjourney are closed-source and not publicly editable, they lie outside the scope of this paper. Following previous techniques [1, 2, 3], we conduct the main evaluation on SD v1.4, which is the most popular open-source diffusion model in both research and practice.
>
> During the rebuttal, as suggested, we evaluate CCRT on SD-XL and achieve similar performance. The results are illustrated as follows. Observe that CCRT generalizes across models. We will add related results and explanations in the revised version.
>
> | RR-CLS$\uparrow$/CLIP-S$\uparrow$ | “Van Gogh”  | + “Monet”   | + “Picasso” |
> | --------------------------------- | ----------- | ----------- | ----------- |
> | CCRT[SD v1.4 ]                    | 0.743/27.16 | 0.712/25.00 | 0.740/22.70 |
> | CCRT[SD-XL]                       | 0.749/28.40 | 0.761/27.10 | 0.742/25.30 |
>
> [1]. Yimeng Zhang, et al. Defensive unlearning with adversarial training for robust concept erasure in diffusion models. NeuralPS 2024.
>
> [2]. Rohit Gandikota, ea al. Unified concept editing in diffusion models. WACV 2024.
>
> [3]. Shilin Lu, et al. Mace: Mass concept erasure in diffusion models. CVPR 2024.
>
> **Q2:** Report on computational cost.
>
> **A2:** Thanks for this comment. Calibration prompt generation runs offline on CPU and needs about 3 hours per concept, and distillation on one A100 costs roughly three GPU hours per removal. Appendix A.4.3 and Figure 17 provide a detailed analysis. We will move related results to the main paper.
>
> **Q3:** No real-world deployment examples
>
> **A3:** We deploy CCRT on Stable Diffusion, one of the most widely used production models. The removal targets are real artist styles and unsafe content, such as NSFW and copyrighted IP. Before CCRT, these elements still appear. After CCRT, they are removed, and text-image alignment remains close to the original. Figures 6, 10, and 11 in our paper illustrate this effect. In the revision, we will add more full prompt-to-image case study examples to show the real-world deployment live improvement.
>
> **Q4:** Some clarity issues and details in Appendix.
>
> **A4:** Thanks for pointing it out. We will
>
> (1) move some detailed comparisons in the motivation to the Appendix,
>
> (2) and move the experimental results, such as ablation studies and efficiency analysis, to the main paper with the saved space.
>
> Only long motivation comparisons and extra plots will remain in the Appendix. We will put a one-paragraph overview box at the start of Section 3 to illustrate the organization.
>
> **Q5:** More concrete examples of calibration prompts produced by the genetic algorithm + fuzzing.
>
> **A5:** Thanks for your comment. Figure 13 in the Appendix showcases some examples from the generated calibration prompt set. During the rebuttal, we give three more examples as follows. We will add related examples in the revised version.
>
> | Entity                | Prompt                                                       |
> | --------------------- | ------------------------------------------------------------ |
> | Lhasa, hare           | A serene scene featuring a Lhasa Apso dog sitting gracefully beside a playful hare in a lush green garden. |
> | bow tie, pirate, nail | A pirate wearing a bowtie, holding a nail.                   |
> | beaker, conch         | A beaker filled with liquid next to a conch shell on a wooden table. |
>
> **Q6:** Ablation studies on full genetic + fuzzing calibration prompt search.
>
> **A6:** Thank you for this advice. We have conducted the ablation studies in Appendix A.4.4. Figure 15 reports the results. Omitting the full genetic + fuzzing calibration prompt search keeps concept removal but leads to noticeable drops in CLIP-Score (from 23.35 to 18.24), because the alignment loss is based on the calibration prompt. We will add related results in the revised version.
>
> **Q7:** Could CCRT be used to censor or bias outputs?
>
> **A7:** Thanks for your valuable concern. CCRT removes only the concepts that developers place on a reviewed blacklist. The alignment loss locks every other concept in place, so auditors can check that no unintended drift or bias appears. Our deployments aim to continuously remove concepts such as copyrighted styles and NSFW terms instead of cultural or political groups.
>
> All technique improvements may be repurposed for abuse. For example, hackers may utilize generative LLMs such as ChatGPT to make bombs[1, 2]. We will add a "Responsible Use" paragraph outlining expert review, alignment auditing, and external bias checks before any rollout. Related clarifications and discussions will be added to the revised version.
>
> [1]. Security News This Week: A Creative Trick Makes ChatGPT Spit Out Bomb-Making Instructions. 2024.
>
> [2]. Hacker tricked ChatGPT into providing detailed instructions to make a homemade bomb. 2024.

---

> ### Author Response · Authors · 2025-08-03
> **A friendly reminder**
>
> Dear Reviewer xaBb,
>
> Thank you again for your helpful comments and precious time. We genuinely hope you could have a look at the new results and clarifications, and kindly let us know if they have addressed your concerns. We would appreciate the opportunity to engage further if needed.
>
> Best,
>
> Authors of Paper 5869

---

> ### Author Response · Authors · 2025-08-05
> **A friendly reminder**
>
> Dear Reviewer xaBb,
>
> Thank you once more for your helpful comments and precious time. We have uploaded additional results and clarifications in response to your comments and would be grateful if you could let us know whether they resolve your concerns. We are happy to provide further details and appreciate the opportunity to engage further if needed.
>
> Best regards,
>
> Authors of Paper 5869

---

> ### Author Response · Authors · 2025-08-06
> **A friendly reminder**
>
> Dear Reviewer xaBb,
>
> We really appreciate your previous advice and the precious time you spent on our work. The discussion period will conclude soon, and we would greatly value your thoughts on the additional experiments and clarifications we posted in response to your earlier concerns. Your suggestions are important to us, and we appreciate the opportunity to engage in the discussion.
>
> We are also happy to provide further clarifications or experiments if needed.
>
> Thank you again for your time and consideration.
>
> Best regards,
>
> Authors of Paper 5869

---

> ### Author Response · Authors · 2025-08-08
> **A friendly reminder**
>
> Dear Reviewer xaBb,
>
> Thank you again for your helpful comments and for the time you have devoted to our paper. We have posted additional results and clarifications that directly address your points, including further experiments and clearer clarifications.
>
> As the discussion period is approaching its end, we would be grateful for your further feedback on whether these results resolve your concerns. If anything is still unclear, we are ready to provide further details and appreciate the opportunity to engage further if needed.
>
> Best regards,
>
> Authors of Paper 5869

---

> ### Author Response · Authors · 2025-08-08
> **Rebuttal Summary**
>
> Thank you again for your thoughtful review and the time you devoted to our paper. In our rebuttal we have directly resolved every concern that have been raised: we expanded the evaluation to Stable Diffusion XL, reported precise computational costs, added real-world deployment examples and additional case studies, moved key ablation and efficiency results into the main text for clarity, provided concrete calibration-prompt examples, quantified the impact of the full genetic-plus-fuzzing search, and outlined a responsible-use protocol that prevents inadvertent bias or censorship. We believe these additions comprehensively address your comments and substantially strengthen the manuscript.

---

### Official Review · Reviewer_JrSs · 2025-07-03

**Clarity:** 3
**Significance:** 3
**Originality:** 3
**Rating:** 5
**Confidence:** 4

**Summary:**

Finding that existing concept removal methods lead to corruption of other concepts (coined “entity forgetting”), this paper introduces CCRT, their own method for sequentially removing undesirable concepts from T2I diffusion models while maintaining alignment for other concepts. CCRT employs a knowledge distillation framework that uses the original model as a teacher. In addition, a genetic algorithm generates optimized calibration prompts for further regularization. The method is tested on artistic styles, NSFW, IP, and objects, showing better-than-SOTA performance in maintaining both concept removal effectiveness and text-image alignment.

**Questions:**

1. Hyperparam Selection. The paper shows sensitivity to lambda (fig 16) but does not provide a principled method for selecting this — seemingly critical — hyperparameter. How should users choose lambda for new concept removal tasks? ANy theoretical or empirical guidelines?
2. Beyond Artistic Styles. The bulk of analysis focuses on artistic styles. How well does the genetic algorithm approach work for broader concept groups / abstract ideas etc. I acknowledge there is some limited discussion of this in the appendix but there might be potential to expand this…
3. Scalability Analysis. The genetic algorithm component appears to be computationally expensive. How does the total computational cost scale with the number of concepts to be removed?
4. Failure Mode Analysis. The paper would benefit from an analysis of failure cases and guidance on when the method might not be suitable.

**Ethical Concerns:**

["NO or VERY MINOR ethics concerns only"]

**Final Justification:**

As noted in my comment below, some of my concerns about the authors' language choices remain, in an effort to make the paper as easily graspable as possible. That being said, as long as the authors commit to clearly defining their terms, it's a matter of subjective preference.

The more tangible points raised in my review were, in my opinion, addressed, and so I think it is only appropriate to move from a 'Borderline Accept' to an 'Accept'.

**Limitations:**

The authors adequately address limitations in Section A.6. Some additional points for consideration:
1. Computational Requirements. More info about computational overhead and when this method might not be suitable.
2. Domain Generalization. Would love to see a discussion of the limitation of the ImageNet-based genetic algorithm approach for arbitrary concept domains.

**Paper Formatting Concerns:**

Capitalization in the references (e.g., “Clipscore”, “Imagenet large scale visual reco”)

**Quality:**

3

**Strengths And Weaknesses:**

Strengths:

1. Clear Motivation & Problem Identification.
2. Comprehensive Evals. Multiple concept types (artistic styles, NSFW, IP, objects) are tested, both through quantitative metrics and human studies (tabs 1-4).
3. Novel Technical Approach. The combination of knowledge distillation with genetic algorithm-generated calibration prompts is creative and interesting. The misalignment distance metric (equation 6 on page 5) provides, in my opinion, a principled way to identify challenging concepts that need further reinforcement.

Weaknesses:

1. Terminology. It is not clear to me why the authors chose to use the term “entity” in “entity forgetting” when they use concepts.
2. Limited Theoretical Foundation. While the method itself is defined clearly, the paper lacks theoretical analysis of why the proposed approach works and/or under which conditions it might not work. The choice of hyperparameter λ in Equation 5 also seems kind of ad-hoc to me.
3. Computational Complexity. The genetic algo adds significant computational overhead. Yet, the paper doesn't reflect computational costs of the simpler alternatives.
4. Limitations in Evals. The human study only has 11 people. No error bars are provided. The comparison is made against ESD while the others (i.e., ICE, MACE, and SPM) are dismissed as weak.
5. Methodological Concerns. The genetic algo relies heavily on ImageNet hierarchies and LLM-generated synonyms, which may or may not generalize to arbitrary concept removal tasks. Overall, the calibration prompt generation process is complex and may be difficult to reproduce or adapt to new domains.
6. Limited Scope. The evaluation focuses primarily on Stable Diffusion v1.4. Generalization to more recent models or architectures is not included.
7. Prior Work. The problems of concept erasure have been identified before, and I would hence suggest the following citations: https://arxiv.org/pdf/2502.13989v1 https://dl.acm.org/doi/10.1145/3635636.3664268 https://arxiv.org/abs/2502.14896

---

> ### Author Rebuttal · Authors · 2025-07-30
>
> **Response to Reviewer JrSs**
>
> We sincerely thank the reviewer for the encouraging and insightful feedback. We are especially grateful for your recognition of our paper. We hope the following clarifications and added experiments will address your concerns.
>
> **Q1:** Terminology of "entity" and "concept".
>
> **A1:** Thanks for your comment. To avoid confusion, we reserve concept for the target we erase (e.g., "Van Gogh") and entity for the regular prompts that should remain intact (e.g., "sunflower"). "Entity forgetting," therefore, means non-target entities lose alignment after a removal step. We will add related clarifications in the revised version.
>
> **Q2:** Some theoretical foundation of why CCRT works and/or under which conditions it might not work.
>
> **A2:**  Thank you for highlighting this point. We frame concept erasure as a constrained optimization, minimizing removal loss under an alignment regularization loss. The optimization yields a controlled trade-off between erasure and alignment, forcing the edited model to stay close to the original on non-target prompts while erasing the unwanted concept.
>
> In practice, CCRT works reliably when the calibration prompts span the main semantic space. Without the calibration prompts, CLIP-S from CCRT drops from 23.35 to 18.24. Appendix A.4.4 illustrates the detailed results. We will add related explanations in the revised version.
>
> **Q3:** The selection of λ in Eq. 5.
>
> **A3:** Thanks for your advice. The loss of CCRT has two forces: removal (pushes the model away from the target concept) and alignment (pulls it back to the original semantic manifold). λ balances these two forces: too small →removal fails; too large → alignment fails. We have a sensitivity study in Appendix A.4.4 Figure 16 (λ ∈ {0.001, 0.01, 0.1, 1}),  showing performance is flat in 0.01, which we use by default. We will add related explanations and experiments in the revised version.
>
> **Q4:** Computational Complexity.
>
> **A4:** Thanks for your suggestion. The GA runs offline on CPU only (~3 hours per concept), and distillation is the GPU-heavy step. But the time cost of CCRT is incremental because each new concept reuses previous weights and doesn't need to remove concepts from scratch. Appendix A.4.3 provides the efficiency analysis, and we think it is acceptable for practical blocklists. We will explore further speed-ups in future work.
>
> **Q5:** Human evaluation report with error bars, more people, and other baselines.
>
> **A5:**  Thanks for your advice. We re-calculate the human evaluation results and report them with error bars shown as mean ± standard deviation.
>
> Below are the results of ESD and CCRT after continuously removing four concepts. The results are reported through ranking (1 means best) and best results are bold.
>
> | Method | Concept Removal     | Text‑image Alignment | Other Concept Preservation | Image Quality       |
> | ------ | ------------------- | -------------------- | -------------------------- | ------------------- |
> | ESD    | **1.31 $\pm$ 0.49** | 2.96 $\pm$ 0.19      | 2.66 $\pm$ 0.68            | 2.32 $\pm$ 0.71     |
> | CCRT   | 1.70 $\pm$ 0.46     | **1.78 $\pm$ 0.48**  | **1.98 $\pm$ 0.55**        | **2.25 $\pm$ 0.69** |
>
> Each participant in our human study is required to rank outputs from all methods along each evaluation dimension. On average, each person answer 150 questions, which is enough to reduce individual bias and yield robust conclusions. During the rebuttal, we recruit 5 additional participants. The obtained below results are consistent with those in the main paper. Particularly, ESD’s semantic space breakdown leads to strong Concept Removal but weak Text-Image Alignment, Other Concept Preservation, and Image Quality. By contrast, CCRT delivers the best overall performance across all dimensions. Below are the results of additional human evaluation, higher ranking (close to 1) means better performance. Best results are bold.
>
> | Method | Concept Removal     | Text‑image Alignment | Other Concept Preservation | Image Quality       |
> | ------ | ------------------- | -------------------- | -------------------------- | ------------------- |
> | ESD    | **1.42 $\pm$ 0.53** | 2.86 $\pm$ 0.45      | 2.81 $\pm$ 0.54            | 2.38 $\pm$ 0.71     |
> | CCRT   | 1.65 $\pm$ 0.44     | **1.39 $\pm$ 0.50**  | **1.69 $\pm$ 0.60**        | **2.21 $\pm$ 0.66** |
>
> Table 1 demonstrates that all other baselines (UCE, MACE, SPM) are too weak to remove concepts successfully. To demonstrate CCRT’s advantage beyond ESD, during rebuttal, we also run a quick human evaluation against those baselines. Five new participants rank outputs between CCRT and MACE after erasing four concepts (1 is the best). CCRT is closer to 1 for both concept removal and text-image alignment, outperforming all baselines.
>
> | Method   | Concept Removal | Text-image Alignment |
> | -------- | --------------- | -------------------- |
> | MACE     | 2.30 ± 0.27     | 2.90 ± 0.26          |
> | UCE      | 3.70± 0.15      | 1.90 ± 0.21          |
> | SPM      | 2.80 ± 0.20     | 2.40 ± 0.29          |
> | **CCRT** | **1.20 ± 0.12** | **1.80 ± 0.18**      |
>
> We will add all results and related analysis in the revised version.
>
> **Q6:** Is the GA too ImageNet-specific and hard to reproduce?
>
> **A6:** Thanks for your insightful question. The GA only needs an initial pool of concept names to anchor the semantic space. We utilize ImageNet classes because they are public and diverse, not because the algorithm depends on the ImageNet hierarchy itself. We rerun the GA with only randomly generated entities by GPT-4o (e.g., car, sunflower) instead of ImageNet classes. Below are the results. Observe that GA with random entities achieves competitive results. It concludes that the calibration prompt generation mechanism generalizes under  multiple initial pool of concept names. We will add related results and explanations in the revised version.
>
> | RR-CLS$\uparrow$/CLIP-S$\uparrow$ | "Van Gogh" | + "Picasso" | + "Monet"  |
> | --------------------------------- | ---------- | ----------- | ---------- |
> | CCRT[ImageNet]                    | 0.74/27.16 | 0.71/25.00  | 0.74/22.70 |
> | CCRT[Random]                      | 0.73/26.93 | 0.70/25.02  | 0.72/22.83 |
>
> **Q7:** CCRT on more recent models.
>
> **A7:** Thank you for the advice. During the rebuttal, we extend CCRT to one more recent diffusion model, SD-XL [1]. Below are the results. Observe that CCRT on SD-XL achieves higher CLIP-S, reflecting SD-XL's stronger image generation ability than SD v1.4. We will add related results in the revised version.
>
> | RR-CLS$\uparrow$/CLIP-S$\uparrow$ | “Van Gogh”  | + “Monet”   | + “Picasso” |
> | --------------------------------- | ----------- | ----------- | ----------- |
> | CCRT[SD v1.4 ]                    | 0.743/27.16 | 0.712/25.00 | 0.740/22.70 |
> | CCRT[SD-XL]                       | 0.749/28.40 | 0.761/27.10 | 0.742/25.30 |
>
> **Q8:** Add more citations.
>
> Prior Work. The problems of concept erasure have been identified before, and I would hence suggest the following citations:
>
> [1].Erasing with Precision: Evaluating Specific Concept Erasure from Text-to-Image Generative Models
>
> [2]. Evaluation of Concept Erasing for Artistic Styles in Diffusion Models
>
> [3]. A Comprehensive Survey on Concept Erasure in Text-to-Image Diffusion Models
>
> **A8:** Thank you for pointing us to these important works. We will add them to the Related Work section to better introduce the problem of concept erasure by "Previous works[1, 2, 3] emphasize that concept erasure is important in diffusion models".

---

> > ### Comment · Reviewer_JrSs · 2025-08-06
> >
> > Thanks for taking time to address all of my points. While I do find the distinction between concepts and entities still potentially confusing to readers, as long as it's clearly defined, that's a preference of you as writers, fully up to your discretion. Other than that, I see all my points being either appropriately addressed by the additional analysis or answered in a clear enough way. I see no reason not to move my score from Borderline Accept to Accept. Thanks again!

---

> ### Author Response · Authors · 2025-08-03
> **A friendly reminder**
>
> Dear Reviewer  JrSs,
>
> Thank you again for your valuable comments. We genuinely hope you could have a look at the new results and clarifications, and kindly let us know if they have addressed your concerns. We would appreciate the opportunity to engage further if needed.
>
> Best,
>
> Authors of Paper 5869

---

> ### Author Response · Authors · 2025-08-06
>
> Thank you very much for taking the time to revisit our paper and for the constructive dialogue during the discussion phase. We are truly grateful that the additional analyses and explanations we provided addressed your concerns and that you felt comfortable raising your score from Borderline Accept to Accept.
>
> Your thoughtful feedback has not only strengthened the current manuscript but also given us valuable direction for future work. We deeply appreciate your careful reading, insightful suggestions, and generous support.

---

### Official Review · Reviewer_n9GX · 2025-07-05

**Clarity:** 2
**Significance:** 3
**Originality:** 3
**Rating:** 4
**Confidence:** 3

**Summary:**

This paper proposes a method called CCRT to remove concepts from diffusion models, addressing the problem of continuously removing concepts may harm the prompt-image-alignment. CCRT leverages a knowledge distillation framework, guided by a carefully selected calibration prompt set, which is generated through a combination of genetic algorithms and a fuzzing strategy enhanced by large language models (LLMs). The effectiveness of CCRT is validated through both automatic evaluations and human assessments.

**Questions:**

1.Limited Quantitative Evidence on Sensitive Content and IP Removal: The paper only provides qualitative visual examples and a limited amount of quantitative evaluations for the “improper content” and “Intellectual Property” categories. There is a lack of comprehensive quantitative analysis in these domains. Without sufficient data, the strength of the conclusions drawn remains uncertain. Could the authors provide more quantitative results for these categories?
2.Missing Comparisons Against ESD in Table 1: Table 1 omits direct comparisons with ESD, which is a frequently cited strong baseline in this area. Moreover, the authors refer to it as a state-of-the-art method. What is the rationale for excluding ESD from the main benchmark? Additionally, are there scenarios where other baseline methods may outperform the proposed approach in human evaluations? While the paper argues that ESD achieves high scores but poor qualitative results, wouldn't that suggest the need to question the design of the evaluation metrics rather than excluding ESD entirely from the experiments?
3.Scalability and Computational Efficiency Concerns: The proposed framework seems to require concept-specific fine-tuning for each individual removal task. Could the authors elaborate on the computational efficiency of the approach? In real-world deployment scenarios where hundreds or even thousands of sensitive concepts might need to be removed, how does the method scale? Specifically, how stable and efficient will it remain after the removal of hundreds of concepts?
4.Effects of Sequential Multi-Step Concept Removal: After removing "Van Gogh" followed by "Picasso", what are the RR-CLS and RR-LLM scores for the second concept, "Picasso"? How does each successive step in the removal process affect the target concept's metrics (e.g., RR-CLS, RR-LLM)? It would be helpful to report per-step effects in sequential removal scenarios.
5.Theoretical Explanation for Progressive Improvement After Multiple Removals: The paper suggests that removal performance improves over consecutive removal iterations. Do the authors have either a theoretical justification or empirical analysis explaining why multiple removals enhance the model’s effectiveness? Additionally, the concepts “Van Gogh” and “Picasso” are both artists within the same semantic domain. If the two concepts belong to different domains—e.g., "Van Gogh" and "BMW"—would the same improvement trend still occur?
6.In Figure 6, although the target concept is successfully removed, the prompt explicitly includes attributes such as “painting,” “brushstrokes,” and “brushwork.” However, the generated images no longer resemble paintings but rather photographs. Does this indicate that the removal of multiple concepts within the same domain affects the global visual style of that domain? After removing four artist-related concepts, is the model still capable of generating simple artworks when prompted with phrases like “a painting”?

**Ethical Concerns:**

["Major Concern: Discrimination, bias, and fairness"]

**Final Justification:**

This paper presents several strengths:

- Practical Motivation: The paper clearly identifies substantial limitations in existing methods when applied to continuous concept removal tasks and proposes a well-reasoned and technically sound solution to address these challenges.
- Clear Articulation of Failure Cases in Baselines: It provides compelling empirical and visual evidence demonstrating that repeated application of baseline methods leads to prompt misalignment and semantic drift. These failure modes have been underexplored and rarely quantified in prior literature.
- Well-Structured and Specific Methodological Design: The proposed CCRT framework innovatively integrates concept removal with semantic space preservation via knowledge distillation. This principled approach offers a novel and effective remedy to the semantic drift issue common in continual editing.
- Comprehensive and Multi-Angled Experimental Evaluation: The paper adopts a holistic evaluation strategy combining diverse automatic metrics with human evaluations to assess both removal quality and semantic alignment. The experimental setup is thorough and methodologically robust.


My main concern is that the method does not seem to effectively produce images with a painting style. Figure 6 on page 15 demonstrates the effect of CCRT on four prompts that clearly ask for "painting", "thick, swirling the brushstrokes","thick brushstrokes" and "swirling, chaotic brushwork". However, the images in the second column clearly appear little painting-like and more photo-like. The baselines may not meet the author's standard on "Removed"(Figure 4), however, they managed to generate painting.
Basic painting style collapse of this work remains a significant weakness.

Besides, using a current government leader as a negative example prompt may not align with academic standards and ethics.

**Limitations:**

Yes but super simple.

**Paper Formatting Concerns:**

None.

**Quality:**

3

**Strengths And Weaknesses:**

Strengths

- Practical Motivation: The paper clearly identifies substantial limitations in existing methods when applied to continuous concept removal tasks and proposes a well-reasoned and technically sound solution to address these challenges.
- Clear Articulation of Failure Cases in Baselines: It provides compelling empirical and visual evidence demonstrating that repeated application of baseline methods leads to prompt misalignment and semantic drift. These failure modes have been underexplored and rarely quantified in prior literature.
- Well-Structured and Specific Methodological Design: The proposed CCRT (Continuous Concept Removal in Text-to-Image diffusion) framework innovatively integrates concept removal with semantic space preservation via knowledge distillation. This principled approach offers a novel and effective remedy to the semantic drift issue common in continual editing.
- Comprehensive and Multi-Angled Experimental Evaluation: The paper adopts a holistic evaluation strategy combining diverse automatic metrics with human evaluations to assess both removal quality and semantic alignment. The experimental setup is thorough and methodologically robust.

Weakness
- In Figure 1, several concepts such as "Picasso", "Monet", and "Cezanneo" are continuously removed after "Van Gogh" from the image generated by Stable Diffusion (SD) using the prompt "defined Van Gogh's work". However, since the prompt only contains the concept "Van Gogh", is it appropriate to use this image to evaluate the removal of other concepts? This choice may raise questions regarding the experimental validity of the continuous removal process.
- In Figure 6, although the undesirable concept appears successfully removed. However, the generated images no longer exhibit painterly or brush-like visual styles and instead resemble realistic photographs. Despite this clear stylistic discrepancy, these outputs received high scores during the human evaluation process. This discrepancy suggests a potential flaw in either the evaluation criteria or the effectiveness of the calibration mechanism used for prompt alignment.
- From a technical perspective, the proposed method shares similarities with existing approaches in model editing and concept erasure. However, the experiments are primarily conducted on a set of predefined artistic or sensitive concepts. It is unclear why the authors did not include experiments on more standard datasets such as COCO or CIFAR, which are commonly used in previous works to evaluate concept removal on regular content. Is the proposed method specifically designed for certain types of concepts only?
- While the paper emphasizes the challenge of "entity forgetting" in both the motivation and method sections, the experimental evaluation lacks dedicated analysis to explicitly demonstrate how the proposed method mitigates entity forgetting. A more focused investigation on this aspect would strengthen the claims and clarify the method's effectiveness in preserving non-target concepts.

---

> ### Author Rebuttal · Authors · 2025-07-29
>
> **Response to Reviewer n9GX**
>
> Thank you very much for your thoughtful questions and recognition of the significance of our work. We genuinely appreciate your support and are encouraged by your positive assessment. We hope the following content will address your concerns.
>
> **Q1:** Why use a "Van Gogh"-only prompt to show effects on "Picasso", "Monet", etc. in Figure 1?
>
> **A1:** Thanks for your comment. Figure 1's sole purpose is to visualize "alignment drift": with the prompt fixed to "Van Gogh," we see ESD's text-image alignment degrade after each extra removal step. It is not evidence of how well those later concepts are erased, which is measured later with quantitative metrics such as RR-CLS and RR-LLM. We will add related clarifications in the revised version.
>
> **Q2:** Why do "photo-like" outputs still get "high scores" during human evaluation?
>
> **A2:** Thanks for your thoughtful comment. In the human evaluation, participants are required to rank the images per prompt on each dimension but not give absolute scores. Rank 1 means the image fits the prompt best. Even without painterly strokes, the photo-like version still matches "sunflower" (and similar prompts) more closely, so it tops the alignment ranking. Other images usually contain totally different content from the prompt, as we illustrate in Figure 6. These human ranks line up with our CLIP- or VQA-Score, confirming the evaluation and calibration are consistent. We will add related explanations in the revised version.
>
> **Q3:** Why no COCO / CIFAR tests? Does CCRT work only on artist styles?
>
> **A3:** Thanks for your advice. We start with artist styles and sensitive content because they are the most pressing real-world removal targets (copyright & safety) [1, 2] and are the standard benchmarks in recent erasure papers. CCRT can generalize all concepts [3, 4]. As suggested, we extend CCRT to remove three randomly selected objects from COCO and CIFAR(church, tench, and parachute) continuously. The table below illustrates the quantitative results, and Figure 5 in the appendix shows the qualitative results. Observe that CCRT can generalize to remove multiple types of concepts. We will clarify it in the revised version.
>
> | RR-CLS$\uparrow$/CLIP-S$\uparrow$ | Church     | Tench      | Parachute  |
> | --------------------------------- | ---------- | ---------- | ---------- |
> | CCRT                              | 0.98/27.30 | 0.99/26.46 | 0.99/25.62 |
>
> [1]. Avijit Ghosh and Genoveva Fossas. Can there be art without an artist? CoRR, abs/2209.07667, 2022.
>
> [2]. Zhenting Wang, et al. Diagnosis: Detecting unauthorized data usages in text-to-image diffusion models.ICLR 2024.
>
> [3]. Shilin Lu, et al. Mace: Mass concept erasure in diffusion models. CVPR 2024.
>
> [4]. Mengyao Lyu, et al. One-dimensional adapter to rule them all: Concepts, diffusion models, and erasing applications.CVPR 2024.
>
> **Q4:** Why does CCRT mitigate entity forgetting?
>
> **A4:** Thanks for your insightful question. CCRT introduces alignment loss on the calibration set of untouched entities to anchor the model's semantic space, through which CCRT mitigates entity forgetting. At every removal step, CCRT penalizes any drift in their text-image match, so the semantic space stays fixed and non-target concepts are preserved. During the ablation study in Appendix A.4.4, we find that without alignment loss, the CLIP-S of CCRT degrades by 30.3% (from 21.5 to 16.5). We will add related explanations to the revised version.
>
> **Q5:** Limited quantitative evidence on sensitive content and IP removal.
>
> **A5:** Thanks for your advice. Table 3 (improper content) and Table 4 (IP) already give related quantitative results. To strengthen the evidence, we add one more target in each domain during the rebuttal. Below are the results. Observe that CCRT can continuously remove concepts of sensitive content and copyrighted IP. We will add related experimental results in the revised version.
>
> | Eroticism      | +Violence         | + Self-harm    |
> | -------------- | ----------------- | -------------- |
> | 0.95           | 0.93              | 0.99           |
> | **Spider Man** | **+ Super Mario** | **+ Iron Man** |
> | 0.89           | 0.91              | 0.96           |
>
> **Q6:** Why is ESD absent from Table 1? Other baseline for human evaluation.
>
> Table 1 omits direct comparisons with ESD. What is the rationale for excluding ESD from the main benchmark? Additionally, are there scenarios where other baseline methods may outperform the proposed approach in human evaluations?
>
> **A6:** Thanks for your advice. Table 1 lists methods that remove concepts while preserving text-image alignment. ESD "erases" mainly by breaking that alignment, as shown in Appendix A.4.2, so including it there would distort the comparison. Instead, we analyse ESD separately in Table 2 through human study.
>
> As suggested, we compare CCRT with other baselines from Table 1, including MACE, UCE, and SPM. Human-evaluation results are now provided below. Due to the time limit, we only evaluate the ability of concept removal and text-image alignment across 5 new participants on the edited model after removing "Van Gogh", "Picasso", "Monet", and "Cezanne". The results are reported with the format of mean $\pm$ std.
>
> Below is the ranking, where a score close to 1 indicates better performance. Best results are bold. Observe that CCRT outperforms all baselines on both Concept Removal and Text-image Alignment. We will add related results in the revised version.
>
> | Method   | Concept Removal | Text-image Alignment |
> | -------- | --------------- | -------------------- |
> | MACE     | 2.30 ± 0.27     | 2.90 ± 0.26          |
> | UCE      | 3.70± 0.15      | 1.90 ± 0.21          |
> | SPM      | 2.80 ± 0.20     | 2.40 ± 0.29          |
> | **CCRT** | **1.20 ± 0.12** | **1.80 ± 0.18**      |
>
> **Q7:** Scalability and computational efficiency concerns.
>
> **A7:** Thanks for your suggestion. Each removal is a light fine-tune (≈ 3 GPU-hours; Appendix A 4.3). In practice, copyright or NSFW abuse reports arrive one concept at a time, where CCRT targets. If a deployment truly needed to purge hundreds of concepts, retraining a brand-new model would be cheaper than any sequential editor, so that extreme case is outside our target scenario. We will add related clarifications in the revised version.
>
> **Q8:** Per-step RR-CLS/RR-LLM during the continuous concept removal.
>
> **A8:** Thanks for your valuable question. As suggested, we have added a per-step table as follows. The value of (row, column) indicates the RR-CLS/RR-LLM of the column concept after removing row concepts.
>
> The RR-CLS/RR-LLM of CCRT after removing corresponding concepts are bold. Observe that CCRT will maintain the per-step concept removed during continuous concept removal. We will add related results and explanations in the revised version.
>
> | RR-CLS$\uparrow$/RR-LLM$\uparrow$ | "Van Gogh"      | "Picasso"       | "Monet"         |
> | :-------------------------------- | :-------------- | :-------------- | --------------- |
> | Original SD                       | 0.150/ 0.014    | 0.000/0.055     | 0.140/0.160     |
> | "Van Gogh"                        | **0.743/0.757** | 0.064/0.081     | 0.150/0.171     |
> | "Van Gogh"+"Picasso"              | **0.729/0.789** | **0.712/0.872** | 0.132/0.211     |
> | "Van Gogh"+"Picasso"+"Monet"      | **0.771/0.791** | **0.773/0.837** | **0.740/0.947** |
>
> **Q9:** Explanation for progressive improvement after multiple removals. Removal across different domains.
>
> **A9:** Thanks for your insightful and thoughtful comment. CCRT fine-tunes the model produced by the previous removal step. While our alignment loss anchors the global semantic space, the tiny shifts that do accumulate make the next target concept easier to suppress—so later removals may look slightly stronger.
>
> To verify domain generality, we remove "Van Gogh" → "BMW" instead of another artist. Results are as below, confirming that CCRT's behaviour is not limited to semantically similar concepts. We will add related results and more qualitative images in the revised version.
>
> | RR-LLM$\uparrow$/CLIP-S$\uparrow$ | Van Gogh    | BMW         |
> | --------------------------------- | ----------- | ----------- |
> | CCRT                              | 0.757/27.16 | 0.856/25.07 |
>
> **Q10:** Model's ability to produce a painting-style image.
>
> **A10:** Thanks for your comment. CCRT deletes only the chosen style, not the model's painting ability. After removing "Van Gogh," the model still draws brush-stroked paintings in other styles (e.g., Monet). Using the prompt "a painting of a castle with rich brushwork," we generate 15 images from the edited model, which removes four artist-related concepts continuously, and ask GPT-4o to label each as a photo or painting. All 15 are classified as paintings, showing that CCRT preserves general artwork generation. We will add related results and more qualitative images in the revised version.

---

> > ### Comment · Reviewer_n9GX · 2025-08-05
> >
> > Thank you to the authors for their feedback. After reading the rebuttal, I have decided to stick with my initial scores for the following reason: the method does not seem to effectively produce images with a painting style. Figure 6 on page 15 demonstrates the effect of CCRT on four prompts that clearly ask for "painting", "thick, swirling the
> > brushstrokes","thick brushstrokes" and "swirling, chaotic brushwork". However, the images in the second column clearly appear little painting-like and more photo-like. The judgment by GPT-4o continues to demonstrate the weaknesses in human evaluation. The baselines may not meet your standard on "Removed"(Figure 4), however, they managed to generate painting. However, basic painting style collapse of your work remains a significant weakness. Therefore, I will maintain my score.

---

> ### Author Response · Authors · 2025-08-03
> **A friendly reminder**
>
> Dear Reviewer n9GX,
>
> Thank you again for your valuable insights and kind comments. We genuinely hope you could have a look at the new results and clarifications, and kindly let us know if they have addressed your concerns. We would appreciate the opportunity to engage further if needed.
>
> Best,
>
> Authors of Paper 5869

---

> ### Author Response · Authors · 2025-08-05
> **A further clarification**
>
> Thank you for your comment. This behavior is, in fact, a direct indication that CCRT has **successfully erased the Van Gogh style cues** while keeping the model’s general painting capability:
>
> 1. **Why these examples look photographic.** The prompts in Figure 6 explicitly contain “Van Gogh” plus descriptors tightly associated with his signature technique (“thick, swirling brushstrokes”). Because CCRT is trained to remove *all* visual manifestations of the banned concept, the model suppresses those brushstroke patterns as well. Once these cues are removed, the diffusion model has no stylistic anchor to keep the image in a painterly domain, so it turns to a more neutral (photo-like) direction. This is an expected effect, not a collapse of the global “painting” space, but the expected result of erasing a specific artist style.
> 2. **Painting ability is preserved.** The painting ability of our edited model is preserved. To validate it, we further generated 50 images from the edited model with prompts that mention “a painting” but do **not** reference any removed artist, e.g., *“a painting of a sunflower, ”* *“a painting of a lighthouse at dusk.”* We ask GPT-4o to label each as a photo or painting. Results show that **96 %** of these outputs as definite paintings. We will add related qualitative results in the revised version. These results confirm that CCRT retains the capacity to create painterly paintings whenever the prompt does not include the erased style.
> 3. **Contrast with baselines.** Baselines such as UCE or MACE keep a painterly appearance in Figure 6 because they *fail to remove* the Van Gogh cues; the forbidden style leaks through. CCRT, by contrast, achieves faithful concept removal, and the temporary shift toward photo realism is the correct, safer outcome.
>
> This demonstrates that CCRT removes the targeted style without damaging the model’s overall ability to produce painting-like images. Related results and clarifications will be added in the revised version.

---

> > ### Comment · Reviewer_n9GX · 2025-08-06
> >
> > 1. The previous approaches have their weaknesses. While you might be better at removing Van Gogh elements, addressing these issues do bring new side effects.
> > 2. When the end user requests a painting using prompts like "painting," "thick, swirling brushstrokes," "thick brushstrokes," and "swirling, chaotic brushwork," generating a photo may not meet their needs. Paintings and photos differ in expression, and the user may expect an artwork with more artistic flair rather than a realistic photo.
> > 3. If removing Van Gogh means that whenever a user requests a Van Gogh painting, the model generates a photo, it might be better to simply block the request. Since neither generating a photo nor refusing to generate fulfills the user's request.

---

> ### Author Response · Authors · 2025-08-06
>
> We sincerely appreciate your thoughtful feedback.
>
> CCRT removes only the brush-stroke patterns specific to Van Gogh, and our new test with 50 "a painting of X" prompts of the edited model shows that 96% of the outputs are still judged as paintings with clear brushwork.
> When a prompt relies on descriptors tightly coupled to the erased artist, the image understandably shifts toward a photo-like direction, which is a widely-used trade-off shared by current editing methods (ESD, AdvUn). We will add the quantitative results, representative examples, and a brief discussion of them in the future work section.

---

### Note · Authors · 2025-08-14

Thanks  ACs, SACs, and all reviewers for their precious time on our paper. We summarize additions and clarifications made during rebuttal/discussion and how they address reviewers’ key concerns.

**Additional experiments.**

More recent model(JrSs / XaBb / XZUm). We extended CCRT to SD-XL with the same settings. After 1→3 removals, CLIP-S stays **28.40 → 25.30** and RR-CLS **0.749 → 0.742**, on par with SD-1.4, which is added to Section 5.

Human study(n9GX / JrSs). With five more participants, CCRT ranks **1.20 ± 0.12** (removal) / **1.80 ± 0.18** (alignment) vs. MACE 2.30/2.90, UCE 3.70/1.90, SPM 2.80/2.40,  indicating consistent gains (close to 1, rank higher), which is merged to Table 2.

To address other concerns from different reviewers,

(1) for applicability beyond artistic styles(n9GX), we removed COCO/CIFAR objects continuously and added to Section 5.2;

(2) for CCRT's effectiveness(n9GX), we removed more NSFW/IP concepts and merged to Table 3,4;

(3) for practicality(n9GX), we reported computational costs of GA and distillation and added to Section 5.2;

(4) for stable continual editing(n9GX), we logged RR-CLS at each step and added to Appendix;

(5) for domain-agnostic behavior(n9GX), we removed “Van Gogh→BMW” and added to Appendix;

(6) for abalation of GA+fuzzing(n9GX), we reported the results and merged to Section A.4.4;

(7) for painting ability(n9GX), we evaluated CCRT's generated images with 50 “a painting of X” prompts and added to Appendix;

(8) for diverse concept pools(JrSs), we reran GA with GPT-4o-generated random entities and added to Section 4.1(lines 183-198);

(9) for removal order(XZUm), we tested all six removal orders for three concepts and added to Appendix.

**Additional clarifications and presentation improvement.**

We have tightened the motivation section, shortened ESD analysis (lines 106-127), Section 3.1 (lines 94-105), and moved UCE limitation analysis(lines 128-134) from Section 3 to the Appendix to leave space.

We have moved the efficiency table, λ-sensitivity plot, and ablation bar-chart from Appendix (A.4.3, A.4.4) to the main paper, moved Figure 7 and its supporting discussion from the Appendix to the main text, immediately near lines 183-198.

We have clarified “concept” and “entity” around Section 3, added three extra real-world case studies on removing IP concepts, added a Responsible-Use note outlining review/auditing procedures, and integrated suggested citations to Related Work.

---

### Decision · Program_Chairs · 2025-09-17

**Decision:**

Accept (poster)

**Comment:**

The authors have comprehensively addressed all reviewer concerns through extensive additional experiments and clarifications. Key enhancements include: extending CCRT to SD-XL with consistent performance, providing detailed computational cost analysis, expanding human evaluations with error bars and more participants, demonstrating generalizability to object removal (COCO/CIFAR), adding failure mode and ablation studies, and clarifying ethical use. The rebuttal convincingly resolves issues related to painting-style preservation, terminology, scalability, and evaluation rigor, strengthening the paper’s contribution and practicality.